## REVIEW ARTICLE

# Role of sleep deprivation in immune-related disease risk and outcomes

Sergio Garbarino [1✉], Paola Lanteri[2], Nicola Luigi Bragazzi[3],
Nicola Magnavita[4,5] & Egeria Scoditti[6]

Modern societies are experiencing an increasing trend of reduced sleep duration, with nocturnal sleeping time below the recommended ranges for health. Epidemiological and laboratory studies have demonstrated detrimental effects of sleep deprivation on health. Sleep exerts an immune-supportive function, promoting host defense against infection and inflammatory insults. Sleep deprivation has been associated with alterations of innate and adaptive immune parameters, leading to a chronic inflammatory state and an increased risk for infectious/inflammatory pathologies, including cardiometabolic, neoplastic, autoimmune and neurodegenerative diseases. Here, we review recent advancements on the immune responses to sleep deprivation as evidenced by experimental and epidemiological studies, the pathophysiology, and the role for the sleep deprivation-induced immune changes in increasing the risk for chronic diseases. Gaps in knowledge and methodological pitfalls still remain. Further understanding of the causal relationship between sleep deprivation and immune deregulation would help to identify individuals at risk for disease and to prevent adverse health outcomes.

Sleep is an active physiological process necessary for life and normally occupying one-third of our lives, playing a fundamental role for physical, mental, and emotional health[1]. Sleep patterns and need are influenced by a complex interplay between chronological age, maturation stage, genetic, behavioral, environmental, and social factors[2–6]. Adults should sleep a minimum of 7 h per night to promote optimal health[7,8].

Besides medical problems including obstructive sleep apnea and insomnia, factors associated mostly with the modern 24/7 society, such as work and social demands, smartphone addiction, and poor diet[9–11], contribute to cause the current phenomenon of chronic sleep deprivation, i.e., sleeping less than the recommended amount or, better to say, the intrinsic sleep need[12].

Sleep deprivation may be categorized as acute or chronic. Acute sleep deprivation refers to no sleep or reduction in the usual total sleep time, usually lasting 1–2 days, with waking time extending beyond the typical 16–18 h. Chronic sleep deprivation is defined by the Third Edition of the International Classification of Sleep Disorders as a disorder characterized by excessive daytime sleepiness caused by routine sleeping less than the amount required for optimal functioning and health maintenance, almost every day for at least 3 months[13].

Population studies reported a stably increasing prevalence of adults sleeping less than 6 h per night over a long period[12,14,15], also affecting children and adolescents[16,17]. Sleep duration

[1] Department of Neuroscience, Rehabilitation, Ophthalmology, Genetics and Maternal/Child Sciences, University of Genoa, 16132 Genoa, Italy. [2] Neurophysiology Unit, Fondazione IRCCS Istituto Neurologico Carlo Besta, Milan, Italy. [3] Laboratory for Industrial and Applied Mathematics (LIAM), Department of Mathematics and Statistics, York University, Toronto, ON M3J 1P3, Canada. [4] Postgraduate School of Occupational Medicine, Università Cattolica del Sacro Cuore, 00168 Rome, Italy. [5] Department of Woman/Child and Public Health, Fondazione Policlinico Universitario Agostino Gemelli IRCCS, 00168 Rome, Italy. [6] National Research Council (CNR), Institute of Clinical Physiology (IFC), 73100 Lecce, Italy. ✉email: sgarbarino.neuro@gmail.com

decline is present not only in high-income and developed countries[18] but also in low-income or racial/ethnic minorities[19], thus representing a worldwide problem.

In addition to fatigue, excessive daytime sleepiness, and impaired cognitive and safety-related performance, sleep deprivation is associated with an increased risk of adverse health outcomes and all-cause mortality[20–24]. Indeed, epidemiological and experimental data support the association of sleep deprivation with the risk of cardiovascular (CV) (hypertension and coronary artery disease) and metabolic (obesity, type 2 diabetes (T2DM)) diseases[24–27]. In the United States, sleep deprivation has been linked to 5 of the top 15 leading causes of death including cardio- and cerebrovascular diseases, accidents, T2DM, and hypertension[28]. Data also point to a role for sleep deprivation in the risk of stroke, cancer, and neurodegenerative diseases (NDDs)[26,29,30]. Sleep deprivation is also associated with psychopathological and psychiatric disorders, including negative mood and mood regulation, psychosis, anxiety, suicidal behavior, and the risk for depression[31–36].

Both too short or too long sleep durations have been found to be associated with adverse health outcomes and all-cause mortality with an U-shaped relationship[37–39]. Although the relation of long sleep duration to adverse health outcomes may be confounded by poor health conditions occurring in older adults[37], the causal association of sleep deprivation with negative health effects is substantiated by experimental evidence providing biological plausibility[24,40,41].

Sleep profoundly affects endocrine, metabolic, and immune pathways, whose dysfunctions play a determinant role in the development and progression of chronic diseases[42–44]. Specifically, in many chronic diseases, a deregulated/exacerbated immune response shifts from repair/regulation towards unresolved inflammatory responses[45].

Regular sleep is crucial for maintaining immune function integrity and favoring a homeostatic immune defense to microbial or inflammatory insults[46,47]. Sleep deprivation may result in deregulated immune responses with increased pro-inflammatory signaling, thus contributing to increase the risk for the onset and/or worsening of infection, as well as inflammation-related chronic diseases.

Here we reviewed the evidence regarding the impact of sleep deprivation on immune-related diseases by discussing the major points as follows: (1) the immune–sleep relationship; (2) the association of sleep deprivation with the development and/or progression of immune-related chronic diseases; and (3) the immune consequences of sleep deprivation and their implications for diseases. Finally, possible measures to reverse sleep deprivation-associated immune changes were discussed.

## Basic immune mechanisms of sleep regulation

The discovery of muramyl peptide, a bacterial cell wall component that is able to activate the immune system and induce the release of sleep-regulatory cytokines, primary regulators of the inflammatory system, provided the first molecular link between the immune system and sleep[48]. Thereafter, other microbial-derived factors such as the endotoxin lipopolysaccharide (LPS)[49], as well as mediators of inflammation, such as the cytokines interleukin (IL)-1 and tumor necrosis factor (TNF)-α, prostaglandins (PGs), growth hormone-releasing hormone (GHRH), and growth factors, were recognized as sleep-regulating factors[50].

Along this line, most animal studies have consistently shown a role in particular for IL-1, TNF-α, and PGD2 in the physiologic, homeostatic non-rapid eye movement (NREM) sleep regulation, so that the inhibition of their biological action resulted in decreased spontaneous NREM sleep, whereas their

administration enhanced NREM sleep amount and intensity, and suppressed rapid eye movement (REM) sleep[51–53]. Moreover, the circulating levels of IL-1, IL-6, TNF-α, and PGD2 are highest during sleep[54]. Their effects are dose- and time-of-day-dependent so that, for instance, low doses of IL-1 enhance NREMS, whereas high doses inhibit sleep[55]. Reciprocal effects may be involved in sleep regulation: for instance, the effects of systemic bacterial products such as LPS may also involve TNF-α[49]. Links exist between IL-1β and GHRH/growth hormone (GH) in promoting sleep so that IL-1 induced GH release via GHRH[56], and hypothalamic γ-aminobutyric acid (GABA)ergic neurons (promoting sleep) are responsive to both GHRH and IL-1β[57]. Instead, anti-inflammatory cytokines, including IL-4, IL-10, and IL-13, inhibited NREM sleep in animal models[58].

Through these substances, the immune system may signal to the brain and interact with other factors involved in sleep regulation such as neurotransmitters (acetylcholine, dopamine, serotonin, norepinephrine, and histamine), neuropeptides (orexin), nucleosides (adenosine), the hormone melatonin, and the hypothalamus-pituitary axis (HPA) axis. Signaling mechanisms to the brain also involve vagal afferents: for instance, vagotomy attenuates intraperitoneal TNF-α-enhanced NREMS responses[59].

Cytokines are produced by a vast array of immune cells, including those resident in the central nervous system (CNS), and non-immune cells, e.g., neurons, astrocytes and microglia, and peripheral tissue cells[60,61]. Cytokines interact with the brain through humoral, neural, and cellular pathways, and form a brain cytokine network (Fig. 1) able to produce cytokines, their receptors, and amplify cytokine signals[50]. Peripheral cytokines reach the brain through different non-exclusive mechanisms, including blood–brain barrier (BBB) disruption[62], penetration of peripheral immune cells, and via afferent nerve fibers, such as the vagus nerve, a bundle of parasympathetic sensory fibers that conveys information from peripheral organs to the CNS[63].

In the CNS, cytokines mediate a multiplicity of immunological and nonimmunologic biological functions[64], such as synaptic scaling, synapse formation and elimination, de novo neurogenesis, neuronal apoptosis, brain development, cortical neuron migration[65], circuit homeostasis and plasticity[66], and cortical neuron migration[65], and complex behaviors, sleep, appetite, aging, learning and memory[65], and mental health status[67,68].

A common experimental finding is that after damage to any brain area, if the animal or human survives, sleep always ensues[69]. Recent evidence indicates that sleep is a self-organizing emergent neuronal/glial network property of any viable network regardless of size or location, whether in vivo or in vitro[53,70–73]. Several sleep-regulatory substances, e.g., TNF, IL-1, nitric oxide, PGs, and adenosine are all produced within local cell circuits in response to cell use[74,75].

From this point of view, TNF-α and IL-1 are closely interconnected and play a similar role in the regulation of sleep[76–81]. IL-1β and TNF-α self-amplify and increase each other's mRNA expression in the brain[82]. In rats, IL-1[83] and TNF-α[84] mRNAs show diurnal variations in different brain areas, with the highest concentrations recorded during increased sleep propensity and peaks occurring at time of sleep period onset in rats and mice[85].

Sleep-like states in mixed cultures of neurons and glia are dependent in part on the IL-1 receptor accessory protein (AcP)[69,86]. In the brain, there is an AcP isoform, neuron-specific (AcPb)[87], whose mRNA levels increase with sleep loss[88,89]. AcPb is anti-inflammatory, whereas AcP is pro-inflammatory[87,88].

TNF signaling promotes sleep, whereas reverse TNF-α signaling (the soluble TNF receptor) promotes waking[90]. The brain production of TNF-α is neuron activity-dependent[91]. Afferent activity into the somatosensory cortex enhances TNF

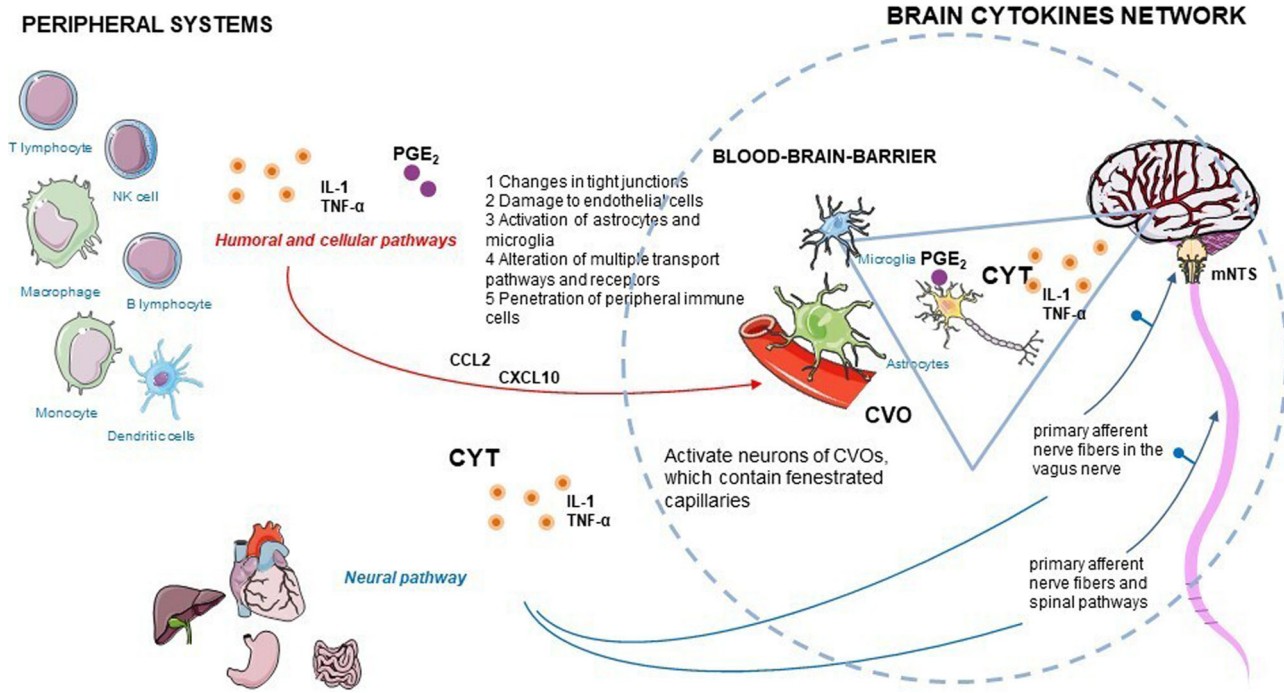

**Fig. 1 Brain cytokines (CYT, orange circles) network and different suggested routes by which peripherally released inflammatory signals can bypass the blood–brain barrier and circumventricular organs (CVOs), and activate the central nervous system: humoral, cellular, and neural pathways.** CCL-2: C-C motif chemokine ligand-2; CXCL-10: C-X-C motif chemokine ligand 10; IL-1: interleukin-1; mNTS: medial nucleus tractus solitarius; PGE$_2$: prostaglandin E2; TNF-α: tumor necrosis factor-α.

expression[92], and in vitro optogenetic stimulation enhances neuronal expression of TNF immunoreactivity[93].

Peripheral immune activation following acute or chronic infection or inflammatory diseases is marketed by altered cytokine concentrations and profiles, and is transmitted to the CNS initiating specific adaptive responses. Among these, a sleep response is induced and has been hypothesized to favor recovery from infection and inflammation, supposedly via the timely functional investment of energy into the energy-consuming immune processes[54,94]. Accordingly, acute mild immune activation enhances NREM sleep and suppresses REM sleep, whereas severe immune response with an upsurge of cytokine levels causes sleep disturbance with the suppression of both NREM and REM sleep[49,95–98]. This sleep change correlates to the course of the host immune response as observed in bacterial and *Trypanosoma* infections[97,99]. Supportively, the increase in NREM sleep was a favorable prognostic factor for rabbits during infectious diseases[96].

Immune regulators also mediate the complex interrelation between sleep and the circadian systems[74]. Circadian rhythms in behavior and physiology are generated by a molecular clockwork located in the suprachiasmatic nucleus, i.e., the master circadian pacemaker, and peripheral tissues, and involving the so-called clock genes (*Clock*, *Bmals*, *Npas2*, *Crys*, *Pers*, *Rors*, and *Reverbs*)[100]. Cytokines, including TNF-α, IL-1β[101,102], and LPS[103–105], suppress the peripheral and hypothalamic expression of core clock genes and clock-controlled genes, resulting in reduced locomotor activity accompanied by prolonged rest time[101].

### Sleep deprivation and immune-related disease outcomes
In the following section, the association between sleep deprivation and risk or outcomes of immune-related disorders, as observed in human studies (mostly observational) and animal experimentations, will be examined. In this context, considering the

sleep–immunity relationship, research has also begun to explore whether and how immune deregulation and inflammation may link sleep deprivation with adverse health outcomes.

**Infection**. A breakdown of host defense against microorganisms has been found in sleep-deprived animals, as shown by the increased mortality after septic insult in sleep-deprived mice compared with control mice[106], or by systemic invasion by opportunistic microorganisms leading to increased morbidity and lethal septicemia in sleep-deprived rats[107]. There is growing evidence associating longer periods of sleep with a substantial reduction in parasitism levels[108] and reduced sleep quality with increased risk of infection and poor infection outcome[109,110]. Accordingly, patients with sleep disorders exhibited a 1.23-fold greater risk of herpes zoster than did the comparison cohort[111]. Furthermore, sleep-deprived humans, as those with habitual short sleep (≤5 h) compared with 7–8 h sleep, are more vulnerable to respiratory infections in cross-sectional and prospective studies[112,113], and after an experimental viral challenge[109,114]. Similarly, compared with long sleep duration (around 7 h), short sleep duration (around 6 h) is associated with an increased risk of common illnesses, including cold, flu, gastroenteritis, and other common infectious diseases, in adolescents[115].

Compared with non-sleep-deprived mice, REM-sleep-deprived mice failed to control *Plasmodium yoelii* infection and, consequently, presented a lower survival rate[110]. This was correlated to an impaired T-cell effector activity, characterized by a reduced differentiation of T-helper cells (Th) into Th1 phenotype and following production of pro-inflammatory cytokines, such as interferon (IFN)-γ and TNF-α, and compromised differentiation into T-follicular helper cells (Tfh), essential to B-cell maturation, which therefore resulted to be reduced[110]. Accordingly, both Maf, a Tfh differentiation factor, and T-bet, a pro-Th1 transcription factor, were reduced in the REM-sleep-deprived group[110]. The combination of REM-sleep deprivation and *P. yoelii* infection

resulted in an additive effect on glucocorticoid synthesis, and chemical inhibition of this exacerbated glucocorticoid synthesis reduced parasitemia, death rate, and restored CD4 T-cell, Tfh, and plasma B-cell differentiation in infected sleep-deprived mice[110], suggesting a role of HPA axis hyperactivation in impairing host immune response under sleep deprivation.

Seep deprivation may exert detrimental effects on sepsis-induced multi-organ damage. Sleep deprivation (3 days) after LPS administration increased the levels of pro-inflammatory cytokines (IL-6 and TNF-α) in the plasma and organs (lung, liver, and kidney), which could be abrogated by subdiaphragmatic vagotomy or splenectomy 14 days prior to LPS administration[116]. Gut microbiota-vagus nerve axis and gut microbiota-spleen axis may play essential roles in post-septic sleep deprivation-induced aggravation of systemic inflammation and multi-organ injuries[116].

Considering the association between sleep deprivation and immune response to infections, vaccination studies allow to assess the impact of sleep and sleep loss on ongoing immune response and the clinical outcome. Studies in which sleep deprivation (one or few nights) was applied to healthy humans during (mostly after) the immunological challenge of vaccination demonstrate that sleep deprivation reduced both the memory and effector phases of the immune response, as indexed by suppressed antigen-specific antibody and Th cell response compared with undisturbed sleep[117].

Congruently, habitual (and hence chronic) short sleep duration (<6 h) compared with longer sleep duration was associated with reduced long-term clinical protection after vaccination against hepatitis B[118]. Sleep deprivation did not exert any impairing effect on mice already immunized[119]. From these studies, it seems that sleep supports—and sleep deprivation impedes—the formation of the immunological memory. Potential mechanisms involved in the beneficial effect of normal sleep on the vaccination response include: (i) the sleep-induced reduction in circulating immune cells that most likely accumulate into lymphatic tissues, increasing the probability to encounter antigens and trigger the immune response; (ii) the sleep-associated profile of inflammatory activation towards Th1 cytokines (increased IL-2, IFN-γ, etc.), which may favor macrophage activation, antigen presentation, and T-cell and B-cell activation; (iii) the effect of sleep stage on the formation of immunological memory through specific immune-active hormones: indeed, during slow wave sleep-rich early sleep, the profile of immune-active hormones, characterized by minimum concentrations of cortisol, endowed with anti-inflammatory activity, and high levels of GH, prolactin, and aldosterone, which support Th1 cell-mediated immunity, may facilitate the mounting of an effective adaptive immune response to a microbial challenge[54].

**Cancer**. Sleep deprivation has increasingly been recognized as a risk factor for impaired anti-tumor response. Epidemiological studies suggest, albeit not consistently[120], a significant association between short sleep duration and the risk for several cancers, including breast, colorectal, and prostate cancer[29,121–123]. Potential mechanisms underlying this association include a shorter duration of nocturnal secretion of melatonin (putatively due to increased light exposure at night)[124], which exerts anti-cancer properties through antimitotic, antioxidant, apoptotic, anti-estrogenic, and anti-angiogenic mechanisms[125]. Melatonin also plays immunomodulatory and anti-inflammatory effects with relevance for its anti-cancer activity, being able to inhibit the pro-inflammatory nuclear factor-κB (NF-κB)/NLRP3 inflammasome pathways, and to support T/B-cell activation and macrophage function[126]. However, besides melatonin, impaired anti-tumor

immune response has been invoked in the sleep deprivation-associated risk for cancer development. A reduced cytotoxic activity of natural killer (NK) cells, which are immune cells with anti-tumor effect, has been reported in 72 h sleep-deprived mice compared with control mice, accompanied by reduced numbers of the cytotoxic cells such as CD8 T cells and NK cells in the tumor microenvironment after chronic sleep deprivation (for 18 h/day during 21 days) in an animal model of experimental pulmonary metastasis[127,128]. In this model, the reduced anti-tumor immunity of sleep-deprived animals was also indexed by the reduced number of antigen-presenting cells (dendritic cells) in the lymph nodes, as well as by the decreased effector CD4 T-cell numbers and corresponding cytokine profile (decreased IFN-γ), resulting in lowered Th1 response of Th cells, i.e., the most effective immune response against tumors. Therefore, an immunosuppressive environment develops with sleep deprivation, which could translate into an early onset and increased growth rate of cancer[128] or increased mortality[129].

An integrated meta-analysis of transcriptomic data showed that circadian rhythm-related genes are downregulated and upregulated in the cortex and hypothalamus samples of mice with sleep deprivation, respectively, with downregulated genes associated with the immune system and upregulated genes associated with oxidative phosphorylation, cancer, and T2DM[130]. Several circadian rhythm-related genes were common to both T2DM and cancer, and seem to associate with malignant transformation and patient outcomes[130].

Hence, although these sleep deprivation-induced immune-mediated mechanisms in cancer warrant further confirmation in humans, the importance of the immune function in the anti-tumor host defense is well recognized[131], thus suggesting that the impaired immune response after sleep deprivation may represent a plausible mediator of the associated increased risk for cancer as described in animal models and in humans.

**Neurodegenerative diseases**. NDDs are aging-related diseases that selectively target different neuron populations in the CNS, and include Alzheimer's disease, multiple sclerosis, Parkinson's disease, Huntington's disease, and amyotrophic lateral sclerosis. One prevailing hypothesis is that altered sleep habits and specifically sleep deprivation may be a consequence and frequently a marker of the disease[132–134]. However, human and animal studies have also suggested a causative or contributing role for sleep deprivation in the development and/or worsening of neurodegenerative processes[132–134].

Potential pathophysiological mechanisms involve, among others, neuro-immune dysregulation. Indeed, a common feature –and a potential therapeutic target- of NDDs is the chronic activation of the immune system, where aspects of peripheral immunity and systemic inflammation integrate with the brain's immune compartment, leading to neuroinflammation and neuronal damage[135]. Neuroinflammation following sleep deprivation has been studied as a pathogenic mechanism potentially mediating the association between sleep deprivation and neurodegenerative processes. Low-grade neuroinflammation as indexed by heightened levels of pro-inflammatory mediators (e.g., TNF-α, IL-1β, and COX-2) and activation of astrocytes and microglia, main immune cells in the brain, was observed in the hippocampus and piriform cortex regions of the brain of chronic sleep-deprived rats along with neurobehavioral alterations (anxiety, learning, and memory impairments)[136]. The sleep deprivation pro-inflammatory milieu was accompanied by oxidative stress in the brain[137] and BBB disruption with consequent increased permeability to blood components[138]. After acute sleep deprivation, there was a significant increased

recruitment of B cells in the mouse brain, which could be important given evidence of B cells involvement in NDDs[139].

Progressive and chronic aggregations of unique proteins in the brain and spinal cord are hallmarks of NDDs[140] and trigger inflammatory responses, gradual loss of physiological functions of the nerve cells, and cell death[141]. Impaired autophagy in humans, a catabolic process of cytoplasmic components, contributes to the aggregation and accumulation of β-amyloid (Aβ), cytoskeleton-related protein τ, and synuclein in neuronal cells and tissues[140]. Sleep plays an important role in the clearance of metabolic waste products accumulated during wakefulness and neural activity. Indeed, the Aβ protein is predominantly cleared from the brain during sleep, possibly through the glymphatic pathway. Congruently, acute and chronic experimental sleep deprivation in animals[142,143] and humans[144] resulted in brain Aβ accumulation and plaque formation, a typical pathological change in Alzheimer's disease process, the most common type of dementia. Imaging studies have revealed that healthy humans with self-reported short sleep were more prone to have cerebral Aβ plaque pathology[145] and disruption of deep sleep (slow wave sleep) increases Aβ in human cerebrospinal fluid (CSF)[146]. Likewise, patients with insomnia present higher CSF levels of Aβ[147].

This pathological Aβ accumulation might reflect disrupted balance of Aβ production and clearance after sleep deprivation. On the one hand, sleep deprivation results in reduced clearance as suggested by clinical studies showing that Aβ levels in CSF are the highest before sleep and the lowest after wakening, whereas Aβ clearance from CSF was impaired by sleep deprivation[148]. Impaired clearance might also derive from disrupted peripheral Aβ transport, as suggested by the sleep deprivation-induced downregulation of low-density lipoprotein receptor-related protein-1 (LRP-1), which promotes Aβ efflux from the brain to the peripheral circulation across the BBB, and elevations of receptor of advanced glycation end products (RAGE), which promotes on the contrary the influx of peripheral Aβ into the brain, thus preventing Aβ clearance[149]. On the other hand, apart from impairing Aβ and τ interstitial fluid clearance, sleep deprivation may also have a role in increasing Aβ and τ exocytosis, thereby increasing CSF Aβ and τ levels[150]. In animals, sleep deprivation also leads to upregulation of β-secretase 1 (BACE-1), the most important enzyme regulating Aβ generation in the brain[142,143,149], thus opening the hypothesis of increased Aβ production by sleep deprivation. Sleep deprivation-induced neuroinflammatory mediators correlate and could lead to disturbed Aβ clearance and stimulated amyloidogenic pathway[143], being pro-inflammatory cytokines able to suppress the expression of LRP-1 and to increase RAGE[151] and BACE-1 levels[152]. Likewise, oxidative stress induced by sleep deprivation may also contribute to the neuroinflammatory burden and the increased expression of BACE-1[153]. Furthermore, patients with insomnia, compared with healthy controls, showed decreased serum levels of neurotrophins, including brain-derived neurotrophic factor (BDNF), proteins especially relevant in neuroplasticity, memory and sleep, and this reduction was significantly related to the insomnia severity[154].

Sleep deprivation is associated with a rapid decline in circulatory melatonin levels, which may be linked to rapid consumption of melatonin as a first-line defense against the sleep deprivation-associated rise in oxidative stress[155]. Melatonin is a potent antioxidant, interacts with BDNF[156], and promotes neurogenesis and inhibits apoptosis[157]. The neuroprotective potential of melatonin can target events leading to Alzheimer's disease development including Aβ pathology, τ hyperphosphorylation, oxidative stress, glutamate excitotoxicity, and calcium dyshomeostasis[150,158]. Accordingly, melatonin treatment could restore the autophagy flux, thereby preventing tauopathy and cognitive decline in Alzheimer's disease mice[159].

Patients with Alzheimer's disease have an increased incidence of sleep-disordered breathing[160]. In addition, sleep-disordered breathing is associated with an increased risk of mild cognitive impairment or dementia and with earlier onset of Alzheimer's disease[161]. Sleep-disordered breathing is also associated with altered levels of Alzheimer's disease biomarkers in CSF, including decreased levels of Aβ and elevated levels of phosphorylated τ[162]. Sleep-disordered breathing possibly via hypoxia, inflammation, and sleep disruption/deprivation could contribute to Alzheimer's disease processes, e.g., increase of Aβ production and aggregation, suppression of glymphatic clearance of Alzheimer's disease pathogenic proteins (τ, Aβ) and oxidative stress, inflammation, and synaptic damage[134,163].

To summarize, the sleep deprivation-associated risk for Alzheimer's disease could be linked to the induction of inflammation in the brain and disorders of systemic innate and adaptive immunity[164]. However, the relationship of sleep deprivation to inflammation in Alzheimer's disease is mostly speculative and needs to be confirmed.

Similar to Aβ in Alzheimer's disease, abnormal levels of α-synuclein are common to Parkinson's disease, the second most common NDDs[165]. Sleep disturbances are not only a common comorbidity in Parkinson's disease, but often precede the onset of classic motor symptoms[166]. The main pathological features of Parkinson's disease are the reduction of dopaminergic neurons in the extrapyramidal nigrostriatal body and the formation of Lewy bodies formed by the aggregation of α-synuclein and its oligomers surrounded by neurofilaments. Due to the degeneration of the dopaminergic neurons, affected people show muscle stiffness, resting tremors, and posture instability; other pathways involved in sleep, cognition, mental abnormalities, and other non-motor symptoms are also affected[167]. Epidemiological studies also suggest that disturbed sleep may increase the risk of Parkinson's disease[168,169]. Such disease-modifying mechanisms may include activation of inflammatory and immune pathways, abnormal proteostasis, changes in glymphatic clearance, and altered modulation of specific sleep neural circuits that may prime further propagation of α-synucleinopathy in the brain[169]. Melatonin could reduce neurotoxin-induced α-synuclein aggregation in mice. Furthermore, melatonin pretreatment reduced neurotoxin-induced loss of axon and dendritic length in dopaminergic neurons through suppression of autophagy activated by CDK5 and α-synuclein aggregation, thereby reducing dyskinesia symptoms in Parkinson's disease animal models[170]. A few reports have shown that melatonin exerts protective effects in several experimental models of Parkinson's disease[171].

However, although animal experimentations suggest a link between sleep deprivation and immune dysfunction in neurodegenerative processes, no human investigations have yet confirmed the mediating role of immune dysregulation in the association between sleep deprivation and risk or outcomes of NDDs.

**Autoimmune diseases**. Sleep disturbances are frequently reported in autoimmune diseases, and immunotherapy in patients with autoimmune pathologies results in sleep improvement[172]. However, knowledge of the immunopathology of autoimmune diseases have disclosed new concepts on the impact of sleep deprivation on autoimmune disease process, showing that sleep deprivation can promote a breakdown of immunologic self-tolerance. Human cohort studies found that non-apnea sleep disorders, including insomnia, were associated with a higher risk of developing autoimmune diseases such as rheumatoid arthritis, ankylosing spondylitis, systemic lupus erythematosus, and systemic sclerosis (adjusted hazard ratio: 1.47, 95% confidence interval (CI) 1.41–1.53)[173] Similarly, in relatives of systemic lupus erythematosus

patients, and hence at increased risk for systemic lupus erythematosus, self-reported short sleep duration (<7 h/night) was associated with transitioning to systemic lupus erythematosus (adjusted odds ratio: 2.0, 95% CI 1.1–4.2), independent of early preclinical features that may influence sleep duration such as prednisone use, depression, chronic fatigue, and vitamin D deficiency[174]. This role of sleep deprivation as a risk factor for autoimmune diseases is corroborated by animal studies. In mice genetically predisposed to develop systemic lupus erythematosus[175], chronic sleep deprivation, applied at an age when animals were yet clinically healthy, caused an early onset of the disease, as indexed by the increased number of antinuclear antibodies, without affecting disease course or severity, according to data on proteinuria, a surrogate marker of autoimmune nephritis, and longevity. Several mechanisms have been postulated to explain the link between sleep deprivation and autoimmune disease risk. Sleep deprivation can accelerate disease development through mechanisms including sleep deprivation-induced increased production of several pro-inflammatory cytokines[44,54], as better discussed below. Indeed, cytokines are synergistically involved in the pathogenesis of autoimmunity, such as IL-6, whose abnormal production results in polyclonal B-cell activation and the occurrence of autoimmune features[176], and IL-17 and the related Th17-cell response[177], which require IL-6 for activation[178] and can cause greater amounts of autoantibody production and immune complex formation, or can intensify chronic inflammation by promoting angiogenesis and recruiting of inflammatory cells at inflammation sites as well as cartilage and bone erosion[179]. Furthermore, experimentally sleep-deprived healthy humans showed impaired suppressive activity of CD4 regulatory T cells (Treg), which normally is highest during the night and lowest in the morning[180]. The suppressive function of Treg towards excessive immune response is an important homeostatic mechanism, whose impairment is implicated in autoimmune disease pathogenesis[181]. Hence, sleep deprivation may not be merely an early symptom or a consequence of an autoimmune disease, but may contribute directly to the pathogenesis increasing the susceptibility to develop an autoimmune disease. More studies are warranted in this field.

**Metabolic and vascular diseases**. Prospective epidemiological evidence associate sleep deprivation (commonly <7 h/night, often <5 h/night) with the incidence of fatal and non-fatal CV outcomes, with a 48% higher risk of coronary heart disease[25], a 15% higher risk of stroke[182], and a 12% increased risk of all-cause mortality[37], which is mainly due to CV causes, according to some authors[183]. In a recent prospective cohort, a low-stable sleep pattern (<5 h sleep/night) during the 4-year follow-up had the highest risk of death and CV events[184]. Short sleep has also been associated with increased subclinical atherosclerotic burden, the dominant underlying cause of CV diseases[185].

In addition, sleep deprivation increases the risk for obesity (about 55% higher risk)[39,186], insulin resistance, T2DM (28% higher risk)[38], and hypertension (21% higher risk)[187], which are powerful and preventable risk factors for CV diseases. Notably, the risk for diabetes attributable to sleep deprivation is comparable to that of other established traditional cardiometabolic risk factors[188], thus underscoring the clinical significance of targeting sleep deprivation in the prevention of cardiometabolic diseases. In contrast with normal nocturnal sleep and in particular NREM sleep characterized by a marked decrease in sympathetic activity, catecholamine plasma levels, and blood pressure, experimental sleep deprivation (acute or chronic) is accompanied by increased sympathetic outflow, with consequent higher blood pressure and heart rate, thus providing a pathogenic link between sleep deprivation and hypertension risk[189–192].

Regarding the influence of sleep deprivation on metabolic pathways, studies support a plausible causal link between sleep deprivation and the risk of overweight and obesity, possibly mediated by the effect of sleep deprivation on circulating levels of hormones (leptin, ghrelin) controlling hunger, satiety and energy balance, besides other factors intervening during sleep deprivation, including physical inactivity and overfeeding[193]. Furthermore, human experimental evidence with chronic sleep deprivation protocol demonstrate that sleep deprivation may alter glucose metabolism[194] and insulin sensitivity[195], thus increasing the risk for obesity and T2DM. The reduction in total body insulin sensitivity observed after sleep deprivation (4.5 h per night for 4 days) in healthy subjects was paralleled by impaired peripheral insulin sensitivity, as demonstrated in subcutaneous fat playing a pivotal role in energy metabolism[195]. Considering a more chronic sleep deprivation, reduced insulin sensitivity was reported in overweight adults after 14 days of experimental sleep deprivation (5.5 h per night) compared with 8.5 h per night of sleep[196], and after habitual curtailment in sleep duration of 1.5 h (<6 h of sleep per night) in healthy young adults with a family history of T2DM[197].

Although the mechanisms that underlie most associations between short sleep duration and adverse cardiometabolic outcomes are not fully understood, potential causative mechanisms involving immune-inflammatory activation have been postulated. It is indeed well established that the subclinical inflammatory status induced by sleep deprivation has pathogenic implications for metabolic and CV risk factors (glucose metabolism, diabetes, hypertension, atherogenic lipid profile, endothelial dysfunction, and coronary calcification) and outcomes (stroke and coronary heart disease)[24]. Accordingly, most of the markers of systemic and cellular inflammation (leukocyte counts and activation state, cytokines, acute-phase proteins, and adipose tissue-derived adipokines) found to be altered after sleep deprivation have been epidemiologically and pathogenically associated with insulin resistance, T2DM, and vascular complications[198]. In fact, inflammation is an early pathogenic process during the development of obesity and insulin resistance[199]. Many adipose tissue-released inflammatory factors with pro-atherogenic and pro-thrombotic actions have also been regarded as a molecular link between obesity and atherosclerotic CV diseases[200]. Furthermore, chronic inflammatory processes are firmly established as central to the development and clinical complications of CV diseases, form the initiation, promotion and progression of atherosclerotic lesions to plaque instability, and the precipitation of thrombosis, the main underlying cause of myocardial infarction or stroke. Most CV risk factors (adiposity, insulin resistance, T2DM, hypertension, and dyslipidemia) act by inducing or intensifying such underlying inflammatory processes that ultimately promote endothelial dysfunction, altered vascular reactivity, innate and adaptive immune system activation, leukocyte infiltration into the vessel wall, and thus atherogenesis[201]. Experimental sleep deprivation leads to endothelial dysfunction, an early marker of atherosclerosis, as indexed by impaired endothelial-dependent vasodilation or increased levels of endothelial adhesion molecules[191].

Among the inflammatory markers, besides being a biomarker of future risk for CV diseases and a predictor of clinical response to statin therapy[202], C-reactive protein (CRP) has been shown to be involved in the immunologic process that triggers vascular remodeling and atherosclerotic plaque deposition[202]. CRP levels lack diurnal rhythm and its liver production is stimulated by cytokines including IL-6 and IL-17, which are upregulated by sleep deprivation[203]. As such, although limited evidence have found an elevation of circulating CRP following sleep deprivation[204], CRP is a prototypical inflammatory factor with

the potential to mark and—to some extent mediate—CV risk following sleep deprivation. Congruently, elevated and sustained plasma levels of CRP have been observed in healthy humans after prolonged sleep deprivation (5 or 10 nights), in concomitance with increased heart rate[190,203], lymphocyte pro-inflammatory activation, and production of cytokines (e.g., IL-1, IL-6, and IL-17)[203]. Similarly, the increase in blood pressure and heart rate observed after acute total sleep deprivation (40 h) was accompanied and even preceded by impaired vasodilation and by increased levels of IL-6 and markers of endothelial dysfunction and activation, such as cellular adhesion molecules (E-selectin, ICAM-1, etc.)[191]. The sleep deprivation pro-atherogenic effect in animal model of sleep fragmentation is mediated, at least in part, by reduced hypothalamic release of hypocretin (i.e., orexin), a wake-inducing neuropeptide, which limits the production of leukocytes (monocytes and neutrophils) and atherosclerosis development, and has been inversely associated with the risk of myocardial infarction, heart failure, and obesity[205]. The activation of the sympathetic nervous system (SNS) may be another mechanism for the inflammatory link between sleep loss and atherosclerotic CV disease, because such activation increases the bone marrow release of progenitor cells, the production of innate immune cells (monocytes), and the levels of inflammatory cytokines, and triggers endothelial dysfunction, thereby leading to systemic and vascular inflammation and atherosclerosis[206,207]. Playing a key role in instigating inflammatory responses and promoting atherosclerosis[208], the sleep deprivation-associated oxidative stress may also contribute to CV risk. It has also been hypothesized a role for melatonin suppression following sleep deprivation in the vascular impairment associated with sleep deprivation, given that melatonin inhibits oxidative stress and cytokine production by immune and vascular cells, and represses atherosclerotic lesion formation in vivo[209].

Therefore, a significant and consistent association exists between sleep deprivation and cardiometabolic risk and clinical outcomes, with several plausible immune-mediated causative mechanisms explaining this association.

### Immune mechanisms linking sleep deprivation and diseases

As shown above, sleep deprivation has been found to alter inflammatory immune processes via multiple pathways, which could lead to increased susceptibility to chronic inflammatory diseases (Fig. 2). Most of the current knowledge on immune effects of sleep deprivation come from studies using controlled experimental sleep deprivation protocols, among which chronic partial sleep deprivation, lasting 2–15 days, is that mostly resembling the human condition of chronic insufficient sleep.

Some studies have observed that sleep deprivation, compared with regular nocturnal sleep, leads to increased circulating numbers of total leukocytes and specific cell subsets mainly neutrophils, monocytes, B cells, CD4 T cells, and decreased circulating numbers and cytotoxic activity of NK cells[203,210–213]. Other studies, however, found contrasting results, including a decrease in CD4 T cells after sleep deprivation[213,214], probably due to differences in sleep deprivation protocol, sampling methodologies, and other factors. Sleep deprivation has also shown to alter circadian rhythm of circulating leukocytes[215], with higher levels during the night and at awakening and a flattened rhythm[210,212]. Additional findings are suggestive of immune deregulation by sleep deprivation, including a decreased neutrophils phagocytic activity[213], altered lymphocytes adhesion molecule expression[216], and reduced stimulated production of IL-2 and IL-12, which are important for adaptive immunity[211,217].

Experimental sleep deprivation has been reported to affect systemic markers of inflammation, with studies showing increased circulating pro-inflammatory molecules (IL-1, IL-6, CRP, TNF-α, and MCP-1); this associated in some studies with a subsequent homeostatic increase in endogenous inhibitors, including IL-1 receptor antagonist and TNF receptors[203,218–220]. In agreement with experimental sleep deprivation, population studies found a direct independent association between habitual short sleep duration (generally < 5 or 6 h) and elevated circulating pro-inflammatory markers, e.g., acute phase proteins (CRP and IL-6), cytokines (TNF-α, IFN-γ, IL-1, etc.), adhesion molecules, and leukocyte counts[183,221–225]. Furthermore, a reduced NK cell activity[226] and a decline in naive T cells[227], compatible with reduced immune competence, was reported in association with habitual short sleep. Shortening of leukocyte telomere length, a cellular senescence marker linked with inflammation, was also associated with shorter sleep duration[228,229].

The reported elevation of systemic inflammation is clinically relevant, because it is suggested to specifically mediate the increased risk of mortality associated with short sleep[23,230,231] and, as observed, the risk for chronic disease development.

Regarding cellular markers of inflammation, some studies found that the ex-vivo LPS-stimulated production of TNF-α[232,233], IL-1β, and IL-6[203,232–234] by human monocytes increased during sleep deprivation but decreased during regular nocturnal sleep[54,203,232–234]. However, other studies reported a decrease of TNF-α production by activated monocytes after sleep deprivation compared with regular nocturnal sleep[203,235]. These contrasting results need further investigations and may depend on differences in the cytokine sensitivity to different sleep deprivation protocols or sampling methods and time. For instance, it seems that partial acute sleep deprivation increased stimulated monocytic TNF-α production[232,233], whereas more sustained sleep deprivation decreased it[203,235].

Undisturbed sleep is predominantly characterized by a Th1 polarization of Th cells (expressing IFN-γ, IL-2, and TNF-α), and experimental sleep deprivation in humans leads to a shift from a Th1 pattern towards a Th2 pattern (expressing IL-4, IL-5, IL-10, and IL-13)[217,236]. Accordingly, conditions featured by disturbed sleep with specific deficit in slow wave sleep, as observed in elderly people[237], alcoholic[238], and insomnia[239] patients, show a cytokine shift towards Th2. The balance of Th1/Th2 immunity and its shift during sleep deprivation may have crucial implications in anti-microbial and anti-tumor immune responses. Th2 over-activity is known to be involved in some forms of allergic responses, and to increase the susceptibility to infection[240]. Likewise, regarding the anti-tumor immune action, Th1 response supports cytotoxic lymphocytes and tumor cells destruction with the potential of elimination or control of tumor cell growth, so that a type 1 adaptive immune response (increased antigen presentation, IFN-γ signaling, and T-cell receptor signaling) may be associated with an improved survival or prognosis[241,242]. In contrast, Th2 over-response is thought to contribute to tumor development and progression, by limiting cytotoxic T lymphocytes proliferation and by the modulation of other inflammatory cell types[241].

Several cellular and molecular signaling pathways may be involved in mediating the influence of sleep deprivation on immune and inflammatory functions (Fig. 3). Increased oxidative stress markers and/or decreased antioxidant defense have been found after sleep deprivation[243–245]. Sleep shows an antioxidant function, responsible for eliminating reactive oxygen species produced during wakefulness, and contrarily sleep deprivation may cause oxidative stress, which leads to cell senescence, unbalanced local/systemic inflammation, dysmetabolism, and immune derangements[246,247].

Effects of sleep deprivation on the immune response may derive from the activation of the SNS with the corresponding

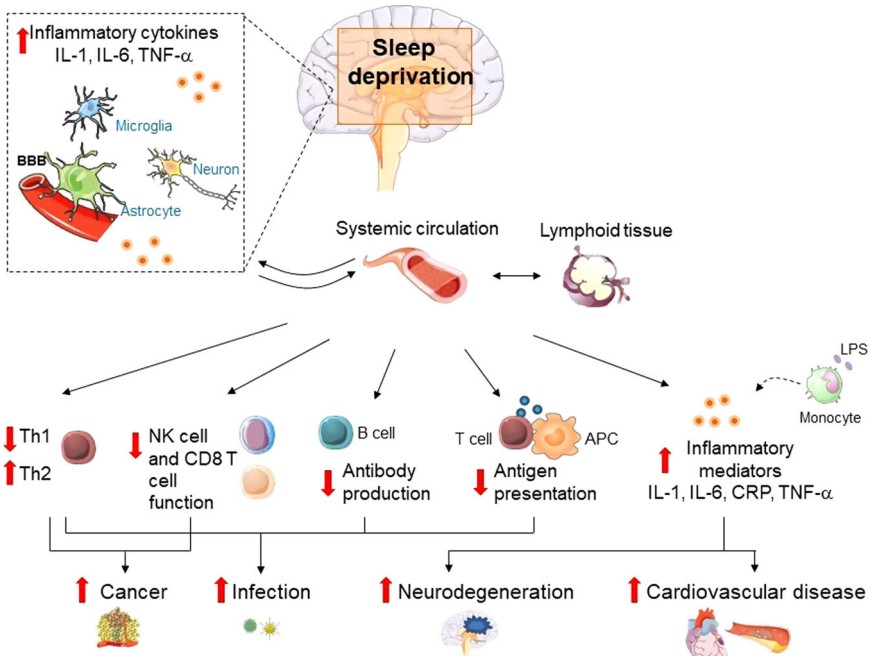

**Fig. 2 Immune consequences of sleep deprivation.** Sleep deprivation, as induced experimentally or in the context of habitual short sleep, has been found to be associated with alterations in the circulating numbers and/or activity of total leukocytes and specific cell subsets, elevation of systemic and tissue (e.g., brain) pro-inflammatory markers including cytokines (e.g., interleukins [IL], tumor necrosis factor [TNF]-α), chemokines and acute phase proteins (such as C reactive Protein [CRP]), altered antigen presentation (reduced dendritic cells, altered pattern of activating cytokines, etc.), lowered Th1 response, higher Th2 response, and reduced antibody production. Furthermore, altered monocytes responsiveness to immunological challenges such as lipopolysaccharide (LPS) may contribute to sleep deprivation-associated immune modulation. Hypothesized links between immune dysregulation by sleep deprivation and the risk for immune-related diseases, such as infectious, cardiovascular, metabolic, and neurodegenerative and neoplastic diseases, are shown. The illustrations were modified from Servier Medical Art (http://smart.servier.com/), licensed under a Creative Common Attribution 3.0 Generic License. APC: antigen-presenting cells.

increase in systemic catecholamines[22,248]. Catecholamines signal to immune cells via adrenergic receptors, which are primarily α- and β-adrenergic in myeloid cells and β-adrenergic in lymphocytes[249]. The immune outcome of the sympathetic signaling is complex, and includes both stimulatory and inhibitory effects depending on cell and receptor types, cell development/ activation states, and local microenvironment[249,250]. Some evidence suggest that β-adrenergic signaling inhibits and α-adrenergic signaling promotes excessive inflammation under endotoxemia[250]. Activation of α-adrenergic signaling in peripheral tissues induces the upregulation of pro-inflammatory cytokines[250,251]. Sympathetic activation also suppresses the transcription of type I IFNs (*IFN-α* and *IFN-β*) genes and interferon response genes, which play a key role in anti-viral immunity[252], and inhibits via β-adrenergic signaling the anti-tumor cytotoxicity of T lymphocytes[253]. In vitro β-adrenergic stimulation repressed Th1 response and stimulated Th2 response, with varying effects found in vivo[249,254]. Although the specific role of SNS activation in the immune phenotype associated with sleep deprivation is not clearly established, data suggest a pro-inflammatory effect of SNS under sleep deprivation. Indeed, chemical sympathectomy has been recently shown to alleviate the inflammatory response following chronic sleep deprivation in mice[255], and both α- and, to a lesser extent, β-adrenergic receptors seem to contribute to the sympathetic regulation of inflammatory responses to sleep deprivation[256].

At the molecular levels, sleep deprivation led to significant gene expression changes in animal tissues[257–259] and human blood monocytes[203,233,260–262], with affected genes mostly related to immune and inflammatory processes (leukocyte function, Th1/ Th2 balance, cytokine regulation, and TLR signaling), oxidative stress, stress response, apoptosis, and circadian system, collectively indicating immune activation and hyperinflammation.

Sleep loss and mistimed sleep also led in the blood transcriptome to alteration and reduction in the circadian rhythmicity of gene expression[261,263], which is an integral part of basic biological processes and homeostasis[264–266].

The activation of the pro-inflammatory NF-κB/Rel family of transcription factors by sleep deprivation, first demonstrated in the late 1990s in mice[267], and subsequently widely confirmed[233,260,261,268–272], is one of the most consistent findings regarding upstream transcriptional regulation. NF-κB induces the expression of genes (e.g., cytokines/chemokines, growth factors, receptors/transporters, enzymes, adhesion molecules) involved in inflammation, immunity, proliferation, and apoptosis[273], circadian clock activity[274], and sleep propensity[275]. Potential signals for NF-κB activation under sleep deprivation include increased adenosine levels, oxidative stress, altered metabolism (adiposity and decreased insulin sensitivity), brain proteins/ metabolites (e.g., Aβ), melatonin suppression[276], circadian clock proteins[277], and catecholamine surge due to increased sympathetic activity[278]. Given the role of NF-κB in the pathophysiology of inflammatory diseases[273], its activation under sleep deprivation may be a common pathway for the risk of morbidity and mortality.

The intestinal microbiota is also affected by sleep loss[279–281], showing indices of dysbiosis (increased Firmicutes:Bacteroidetes ratio; decreased diversity and richness), which may affect the immune system[282], and are similar to those associated with cardiometabolic diseases[45].

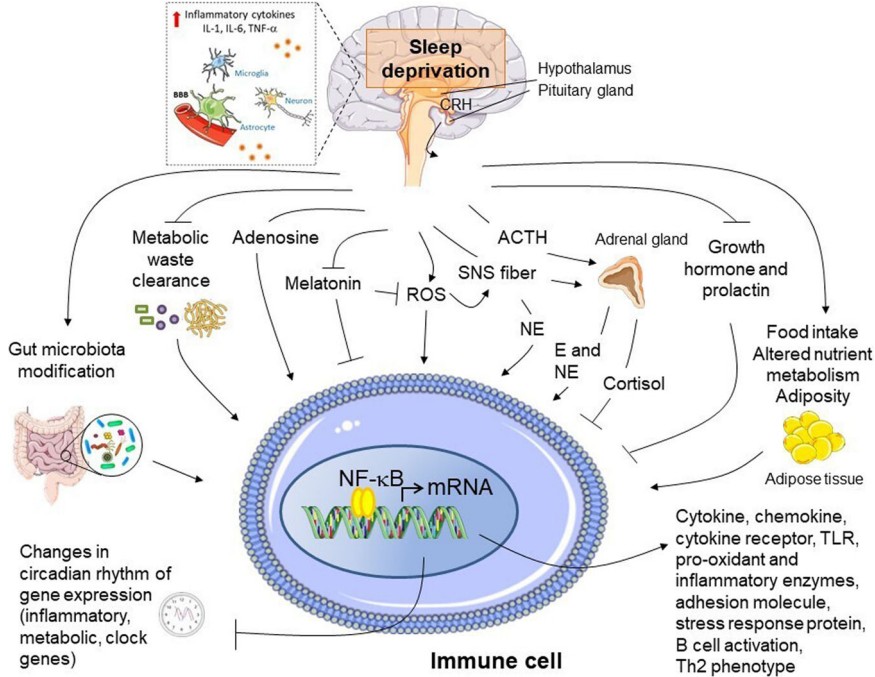

**Fig. 3 Pro-inflammatory molecular pathways induced by sleep deprivation.** A schematic model of potential mechanistic pathways linking sleep deprivation and inflammatory immune activation is depicted. Sleep deprivation is associated with activation of the sympathetic nervous system and release of norepinephrine and epinephrine into the systemic circulation, as well as to some extent with impaired hypothalamus-pituitary axis stimulation. These neuromediators may act along with other potential stimuli accumulated following sleep deprivation including reactive oxygen species (ROS), adenosine, metabolic waste products (e.g., β-amyloid) not cleared during normal sleep, gut microbiota dysbiosis leading to altered local and systemic pattern of metabolic products, as well as with changes in the profile of neuro-endocrine hormones, such as prolactin, growth hormone, and altered circadian rhythm of melatonin secretion. In immune cells located in the brain and the peripheral tissues, these stimuli may in concert trigger inflammatory activation, with release of cytokines, chemokines, acute phase protein, etc. via the recruitment of transcriptional regulators of pro-inflammatory gene expression, mainly nuclear factor (NF)-κB, and disturbing the circadian rhythmicity of gene expression of both clock genes and metabolic, immune and stress response genes (see text for further detail). E: epinephrine; NE: norepinephrine; TLR: Toll-like receptor. Arrows indicate stimulation; lines indicate inhibition. The illustrations were modified from Servier Medical Art (http://smart.servier.com/), licensed under a Creative Common Attribution 3.0 Generic License.

## Countermeasures for sleep deprivation: effect on immune parameters

Although the impact of strategies to improve sleep duration on neurobehavioral performance and alertness after sleep deprivation have been assessed[283–285], sleep deprivation countermeasures to improve immune and inflammatory parameters, and, correspondingly, disease risk and outcomes have been studied to a lesser extent.

Although extension of habitual short sleep did not show to significantly counterbalance the immune consequence of sleep deprivation[286–288], mixed results derive from nighttime recovery sleep following sleep deprivation (Table 1), with limited evidence of effectiveness for specific immune parameters[210,214], and mostly after multiple consecutive nights of 8 h sleep recovery or with an extended nocturnal sleep duration[212,289].

Although daytime napping (<20 min) restores alertness, and mental and physical performance without provoking sleep inertia associated with longer nap[290–292], the effects of a short nap on immune/inflammatory parameters after sleep deprivation have yet to be firmly established. Differently form population studies[293], laboratory studies found immune benefit from nap[218,289,294,295]. Regarding immune-related clinical outcomes, controversy exists, with studies finding no association[296], inverse associations[297,298] or positive association[296], and a J-shaped relationship[299–301] between napping and CV and metabolic diseases or cancer events and mortality. Whether changes in immune parameters could contribute to the associations between napping and immune-related diseases remains unclear.

Among the strategies to recover sleep deprivation-induced immune changes, cognitive behavior therapy improves sleep outcomes in insomnia and lowers cellular and systemic inflammatory markers[302,303], and the risk score composed of CV and metabolic risk factors[304]. This highlights the potential role of targeting sleep in reducing the inflammatory risk and the associated chronic diseases.

## Summary and concluding remarks

Sleep exerts immune-supportive functions and impairments of the immune-inflammatory system are a plausible mechanism mediating the negative health effects of sleep deprivation, and in particular, its role in the risk and outcomes of chronic diseases such as infections, CV, metabolic and autoimmune diseases, NDDs, and cancer. Caution should be exercised in interpreting cellular and molecular outcomes of sleep deprivation in experimental studies conducted till now as a result of an independent effect of sleep deprivation, because other factors may play a role, including extended wakefulness-associated processes, other features of sleep-wakefulness, their temporal and functional segregation or methodologies of sleep manipulation.

Randomized controlled trials assessing the effect of treatment of sleep deprivation on inflammatory immune dysfunction and/ or health outcomes are needed. Knowledge of inflammatory and immunological signatures in response to sleep curtailment may inform not only on the underlying molecular links, but also contribute to refine risk profiles to be used for developing biomarkers of sleep deprivation and sleep disturbance-related health

**Table 1 Main human findings on the effects of recovery sleep on sleep deprivation-induced changes in immune and inflammatory parameters.**

| Subjects (number and age range or mean) | Sleep deprivation protocol | Effect of sleep deprivation on immune parameters compared with baseline | Recovery sleep protocol | Effect of recovery sleep on immune parameters compared with baseline | Effect of recovery sleep on immune parameters compared with sleep deprivation | Reference |
|---|---|---|---|---|---|---|
| Healthy men and women (n = 20, 21–30 yrs) | 64 h TSD | ↑ Granulocytes and monocytes, NK cell activity | 1 Night (h sleep not reported) | ↑ Granulocytes, monocytes; = NK cell activity | = Granulocytes; ↓ monocytes; ↓ NK cell activity | Dinges et al.[214] |
| Healthy men (n = 32, 19–29 yrs) | 2 Nights TSD or 4 nights of REM SD | TSD: ↑ total leukocytes, neutrophils, CD4 T cells; REM SD: ↓ IgA | 3 Nights (8 h sleep/night) | = Total leukocytes, neutrophils; ↑ CD4 T cells ↓ IgA | ND | Ruiz et al.[308] |
| Healthy young men (n = 10, 21–29 yrs) | 1 Night TSD | During SD: ↑ monocytes, lymphocytes, NK cells. The day after SD: ↓ lymphocytes, NK cells | 1 Night (8 h sleep) | = Monocytes, lymphocytes; ↓ NK cells | ND | Born et al.[210] |
| Healthy men (n = 12, mean age 29 yrs) | 40 h TSD | ↑ Plasma E-selectin; ↑ systolic BP, heart rate; plasma norepinephrine; ↓ endothelium-dependent and -independent vasodilation | 1 Night (8 h sleep) | ↑ Plasma ICAM-1, IL-6, norepinephrine | ND | Sauvet et al.[191] |
| Healthy men (n = 31, 18–27 yrs) | 1 Night with 2 h sleep | ↑ Total leukocytes, neutrophils | 1 Night of 8 h sleep or 1 night of 10 h sleep | 8 h Recovery sleep: ↑ leukocytes, neutrophils; 10 h recovery sleep: = leukocytes, neutrophils | 8 h Recovery sleep: = leukocytes, neutrophils; 10 h recovery sleep: ↓ leukocytes, neutrophils | Faraut et al.[289] |
| Healthy men (n = 19, 19–29 yrs) | 5 Nights with 4 h sleep/night | ↓ NK cells; ↑ B cells; ↑ plasma CRP, IL-17, IL-1β, IL-6 (PBMC mRNA); ↓ TNF-α (PBMC protein) | 2 Nights (8 h sleep/night) | = NK cells; = B cells; ↑ CRP, IL-17; = IL-1β, IL-6, TNF-α | ND | van Leeuwen et al.[203] |
| Healthy men (n = 9, 22–27 yrs) | 5 Nights with 4 h sleep/night | ↑ Total leukocytes, monocytes, neutrophils, lymphocytes | 7 Nights (8 h sleep/night) | ↓ Monocytes, lymphocytes; ↑ neutrophils | ND | Lasselin et al.[212] |
| Healthy men and women (n = 24, 36–76 yrs) | 1 Night with 4 h sleep | ↑ IL-6 and TNF-α (PBMC protein) | 1 Night (8 h sleep) | ↑ IL-6 and TNF-α | ND | Irwin et al.[262] |
| Healthy men (n = 10, 22–37 yrs) (Exp. 1) and healthy | 88 h TSD (Exp. 1) and 10 days with 4.2 h | ↑ Plasma CRP | 3 Nights (assessment only in the first recovery day) | ↑ Plasma CRP | ND | Meier-Ewert et al.[190] |

**Table 1 (continued)**

| Subjects (number and age range or mean) | Sleep deprivation protocol | Effect of sleep deprivation on immune parameters compared with baseline | Recovery sleep protocol | Effect of recovery sleep on immune parameters compared with baseline | Effect of recovery sleep on immune parameters compared with sleep deprivation | Reference |
|---|---|---|---|---|---|---|
| men and women (n = 10, 26–38 yrs) (Exp. 2) | sleep/night (Exp. 2) | | | | | |
| Healthy men and women (n = 21, 25–39 yrs and n = 49, 60–84 yrs) | 1 night with 4 h sleep | ↑ IL-6 and TNF-α (PBMC protein) in younger adults | 1 night (8 h sleep) | ↑ IL-6 and TNF-α | ND | Carroll et al.[309] |
| Healthy men and women (n = 30, 18–34 yrs) | 6 Nights with 6 h sleep/night | ↑ Plasma IL-6 | 3 Nights (10 h sleep/night) | = Plasma IL-6 | ↓ Plasma IL-6 | Pejovic et al.[310] |
| Healthy men and women (n = 14, 18–35 yrs) | 5 Nights with 4 h sleep/night, for 3 weeks | ↑ IL-6 (PBMC protein) | 2 Nights (8 h sleep/night), for 3 weeks | ↑ IL-6 (PBMC protein) | ND | Simpson et al.[234] |

BP blood pressure, exp experiment, ICAM-1 intercellular adhesion molecule-1, ND not determined, PBMC peripheral blood mononuclear cells, SD sleep deprivation, TSD total sleep deprivation, VCAM-1 vascular cell adhesion molecule-1, yrs years, ↑ significant increase, ↓ significant decrease, = no significant change.

outcomes, which may also represent potential targets of interventions. Recent metabolomic[305] and transcriptomic[306] studies hold promise in biomarker discovery[306].

These efforts may converge towards a new ground fostering interactions between the sleep research and the medical community to translate scientific knowledge into the clinic, prioritize health issues, and develop strategies and policies for subject risk stratification, to include evidence-based sleep recommendations in guidelines for optimal health and to address sleep hygiene at the individual and the population levels, as a means to prevent the negative health consequences of sleep deprivation. These actions might also foster health literacy and empowerment of individuals to actively better manage their own health and well-being throughout their life course by means of lifestyle, nutritional, and behavioral habits including sleep hygiene[307].

Conclusively, in the perspective of staying healthy in this rapidly changing society, the sleep–immunity relationship raises relevant clinical implications for promoting sleep health and, as evidenced here, for improving or therapeutically controlling inflammatory response by targeting sleep. This may ultimately translate, in the era of preventive medicine, into addressing sleep as a lifestyle approach along with diet and physical activity to benefit overall public health.

**Reporting summary**. Further information on research design is available in the Nature Research Reporting Summary linked to this article.

## Data availability

All data generated or analysed during this study are included in this published article.

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

## Author contributions

E.S. reviewed the literature and wrote the manuscript draft. S.G. and P.L. contributed to writing the manuscript and revised the manuscript draft. N.L.B. and N.M. reviewed the final manuscript.

## Competing interests

The authors declare no competing interests.
