## [Transparent Peer Review File · Communications Biology]

Reviewers' comments:

Reviewer #1 (Remarks to the Author):

The review of Garbarino et al. summarizes research on the bidirectional communication between sleep and the immune system with a particular focus on the health consequences of insufficient sleep.

The review is overall well written and covers an important and current topic that is interesting to a broad, interdisciplinary research community. It could be structured somewhat better, because it touches several broad topics but remains superficial in the discussion of certain areas. To improve this, I would suggest to keep the focus more on the effects of sleep deprivation on disease outcome, which is already an extensive and highly relevant topic that so far has been less covered in publications within the sleep-immune research field. This would also help to reduce the strong overlap with several recent publications that have extensively reviewed basic sleep-immune interactions (Annu Rev Psychol. 2015, PMID: 25061767, Physiol Rev. 2019, PMID: 30920354, Nature Reviews Immun. 2019, PMID: 31289370).

Specific comments:

- Although the manuscript is overall well written, some sentences are quite long or difficult to understand. Some examples are:
- Abstract, line 28f: Why would a "further understanding of the causal relationship between SD ... help ... to improve measures of sleep hygiene"?
- Lines 47-50: "However, in the modern hyper-technological era we are witnessing an impressive increase in chronic sleep problems, that are closely linked to the growing and unstoppable pressure on sleep leading to the development of a constant sleep debt or insufficient sleep, likewise referred to as sleep deprivation (SD).": The phrasing could be simplified.
- Lines 51-57: "Nowadays, apart from medical problems, including obstructive sleep apnea (OSA), insomnia and others, factors associated with the modern 24/7 society such as globalization, societal demand for constant activities, increasing working time and schedule with more frequent shift-work, educational demands, internet explosion and communication/information technology as well as poor diet and drugs, contribute to a misalignment between the biological necessity of good sleep and the intense daily activities and lifestyle, thus changing sleep duration patterns and causing the currently pervasive phenomenon of self-induced chronic SD, that is sleeping less than the recommended amount or, better to say, the intrinsic sleep need⁴." This is an extremely long sentence that could be shortened.
- Lines 83ff: "...sleep hygiene is becoming an increasingly urgent public issue for both the scientific and medical communities in order to promote more awareness and education on sleep at the population and individual levels and hence prevent or attenuate the adverse health outcomes associated with insufficient sleep." What does the part "in order to promote" refer to?
- Lines 373ff: "Indeed, compared with regular nocturnal sleep, acute and mostly sustained SD has been found: a) to alter circulating leukocyte counts with studies reporting increased numbers of total leukocytes and specific cell subsets mainly neutrophils, monocytes, B cells, decreased circulating NK cells, and decreased circulating CD4+ T cells, in comparison with undisturbed sleep which is associated with low leukocyte counts, possibly due to a redistribution of leukocyte subsets to lymphoid organs where they participate in innate and adaptive immunity^{84,87-90};" This sentence is somewhat redundant. In addition, circulating CD4+ T cells were found to be increase rather than decreased by SD.
- The title of the review should more accurately reflect the focus of the manuscript on the role of insufficient sleep in disease risk and outcome.
- Some statements lack references, e.g.

- Lines 42-44
- Lines 97-103
- Lines 343-346
- Lines 356-359
- Lines 562-566

- Lines 113-116: The Introduction ends with the statement "Here, we reviewed the evidence regarding the role of chronic SD and the potential pathogenic mechanisms in the dysregulation of the immune function and the induction of unbalanced pro-inflammatory responses, as a risk factor for the development and/or progression of several immune-related chronic diseases. ", yet the next sections also deal with several other aspects, such as the regulation of sleep by immune products and circadian factors. The relevance of these topics for the main focus of the review should either be clarified more or the review should be more focused on directly relevant topics.

- Lines 196 – 204: The description of sleep stages seems misplaced here. The authors may consider giving a broader introduction to the basics of sleep or remove this description entirely.

- In general, the review would benefit from including also some critical findings with opposite results and from a more in-depth discussion of the findings. These are only some examples:

- Lines 494-497: There is a lot of evidence that β -adrenergic signaling is anti-inflammatory whereas α -adrenergic signaling is pro-inflammatory (see for example PMID: 25703786).

- Lines 595ff: "Moreover, the genomic response to acute total SD is strongly intensified after chronic SD than after sufficient sleep, with more up-regulated genes belonging to IL-6 signaling, inflammatory and defense responses, phagocytosis, response to external stimuli and wounding, suggesting an exacerbated response to stress and challenges and predisposing to negative health outcomes⁵⁶." The conclusion that an up-regulation of genes involved in defense responses and phagocytosis predisposes to negative health outcomes is very speculative. One could likewise argue in the opposite way.

- Lines 1047ff: "In parallel, cognitive behavior therapy reduced the stimulated monocyte production of IL-6 and TNF- α and the expression of pro-inflammatory transcripts." This finding appears to contradict studies that found an increasing effect of sleep on the stimulated production of IL-6 and TNF- α . This could be discussed.

- Lines 521-522: "Normally, cortisol suppresses the pro-inflammatory gene expression (predominantly via mineral corticoid receptor) and upregulates the production of anti-inflammatory cytokines¹⁵³": Please check whether this statement is correct. Many effects of cortisol on pro-inflammatory signaling are mediated via glucocorticoid not mineralocorticoid receptors.

- Lines 688-689: The first sentence is not well connected to the rest of the paragraph.

- Fig. 1: The line indicating an inhibition of T cells is somewhat confusing. From the depiction in the figure, it is not clear whether SD increases or decreases numbers of Th1 and Th2 cells.

- Lines 699-701: "... demonstrate that both the memory phase, indexed by the antigen-specific antibody response, and the effector phase, marketed by T cell number and cytokine production ... ". This statement is not correct. Effector and memory phases are both marked by antigen-specific antibody and T cell responses.

- Table 1: It would be interesting to show the effects of recovery sleep not only compared with changes after SD but also compared with baseline measurements (i.e., before sleep manipulation), to see whether recovery sleep entirely compensates the effects of SD.

- Lines 982-983: The statement "Hence, the ... pro-inflammatory effect of short sleep duration is reversible by increasing habitual sleep duration" should be toned down, because the effect was not significant.

- Lines 1057ff: "Sleep exerts immunosupportive and inflammation-regulatory functions, and impairments of immune inflammatory system is a plausible mechanism mediating the negative health effects of SD, which leads to a response by the host immune defense similar to that of threats (infection or injury).": I don't think it is correct to say that SD leads to an immune response similar to that of infection or injury.

- The manuscript should be carefully screened for grammar errors, e.g., line 87 (plays should read play), line 563 ("are thought the cellular line" should probably read "are thought to be..."), line 628 ("these evidence" should read "this evidence"), line 968 ("Whether SD or habitual short sleep represent...": whether should probably read if), etc.

- The word "subject", is sometimes not used appropriately, e.g. line 1055: "Epidemiological data clearly suggest that increasing numbers of subjects are becoming sleep deprived...".

Reviewer #2 (Remarks to the Author):

Critique of: Manuscript Number: COMMSBIO-21-0088

Corresponding Author: Professor Garbarino

Title: Sleep deprivation as a pathogenic noxa for the immune system: what is the evidence?

This manuscript is a clearly written comprehensive summary of the multiple effects of sleep loss (sleep deprivation [SD]) on immune system related pathologies. In general, citations are appropriate and provide a reasonable guide for readers to go into greater depth. The emphasis on the effects of SD on specific pathologies/syndromes and relating those effects to changes in immune responses is well organized and appropriate. While it may not be unique, it is more focused on the SD-immune links in pathologies than prior reviews in this field, some of which are stellar, e.g. ref 34. Regardless, a few issues were identified. Herein they are organized into major concerns, suggested issue to address, and minor issues.

Major Issues:

1. The brain network of cytokines, including the cells that express them, what stimulates them, their involvement in multiple brain functions, e.g. synaptic scaling, brain development, sleep of course, appetite, mental health status etc. is hardly mentioned. Further, the complexity of the brain cytokine network as it applies to sleep is not mentioned. For instance, within the IL1 and TNF families of molecules there are more anti-inflammatory members than pro-inflammatory members. Further, the effects of IL1 and TNF on sleep are dose-dependent – low doses promote sleep while high doses inhibit sleep – and time of day dependent. Some of the brain cytokine molecules are specific to brain. E.g. the IL1 receptor accessory protein (AcP) has 3 isoforms, one is specific to neurons (called AcPb); AcPb, [not AcP or the soluble form of AcP], upregulates with sleep loss and is required for sleep rebound to occur after sleep loss. AcPb is anti-inflammatory. Other examples illustrating the nuances of the brain cytokine network include the observations that forward TNF signaling (i.e. soluble TNF binding to one of its transmembrane receptors) promotes sleep while reverse TNF signaling (the soluble TNF receptor binding to the transmembrane form of TNF) promotes waking. Finally, both IL1 and TNF have daily rhythms of expression in brain with peaks occurring at ZT0, time of sleep period onset in rats and mice.

2. How systemic cytokines, which are the major focus of human studies of SD-immunity, relate to brain cytokine expression and actions are not developed adequately. E.g. there is a literature indicating that systemic cytokines induce action potentials in vagal afferents which in turn induce brain cytokine mRNA production. Such effects are lost with vagotomy. Similar studies show how bacterial products also are dependent in part on vagal afferents to affect brain cytokine levels. For low grade inflammation, this is likely a major way of influencing the brain cytokine orchestra and many of the SD-linked pathology outcomes such as the autonomic parasympathetic and sympathetic nervous systems influence as was developed in the review. Given the large gap between animal studies, within

which brains can be examined, and human studies, which are mostly restricted to systemic sampling, these mechanisms need to be addressed as they provide validity to the human studies.

3. There are known links between cytokines and circadian rhythm molecular mechanisms. Perhaps most important for this review are those developed by A. Fontana's lab in Zurich where it was shown that TNF and IL1 directly affect clock gene expressions. This could be mentioned in multiple places in the manuscript.

4. In contrast, the authors devote a few paragraphs to the two-process model of sleep and a related triple-drive model. These are mathematical models which are not concerned with molecular mechanisms and compartmentalize sleep regulation into two or three components for which we already know complex interactions, e.g. see reference to Fontana's work above. The models although useful in various ways, are not helpful to link SD to immunity and pathology. Further, the primary outcome measure used to quantify process S, EEG slow wave activity, is regulated independently of sleep duration and does not reflect sleep intensity in multiple conditions, e.g. neonates and the elderly mostly lack EEG SWA, some sleep pills inhibit EEG SWA while increasing sleep, etc. The paragraphs discussing mathematical models could be eliminated without loss meaningful manuscript content..

5. Somewhere in the introduction the simple observations that almost all physiological measures, be it hormone levels, blood CO2 levels, mentation, heart rate, respiratory rate, locomotor activity, etc. are different between sleep and wake states. This coupled with the effects of SD on genome-wide expression of transcripts, which indicate that a very large number of genes are altered by SD, over 50% in some studies, suggest limitations on interpretation of the effects of SD to the extent that the effects observed are the result of sleep loss per se. These broad changes between sleep and wake indicate that few, if any, studies have ever isolated sleep as the independent variable; as such causality is not possible to identify, thereby limiting conclusions.

6. The paragraph (lines 169-176) within which hypotheses dealing with sleep function is insufficiently develop and could be omitted since none of the hypotheses mentioned are universally accepted nor are they relevant to the immune system with the possible exception of the idea the sleep removes toxic biomolecules from the brain. That idea, while very attractive, is mostly based on experimental evidence gathered during states of anesthesia. The ideas of Obal and Krueger (1993) and Kavanau (1995, 1996) positing that sleep is required for preserving, or stabilizing, synaptic plasticity, and is activity-dependent, are not even mentioned even though TNF is a well-characterized brain molecule tied to synaptic scaling. Thus, the only theories of sleep function directly tied to "immune molecules" is not mentioned.

Suggested issues to address:

7. Lines 72-92; It may be interesting to address the multiple findings relating sleep and sleep loss to mental illness which in turn can affect immune function.

8. Line 176; the concept of local use-dependent sleep was first put forward by Obal and Krueger in 1993 and they and others have developed the story in the intervening years, perhaps most convincingly by D. Rector. Some of that work is directly related to the subject of this review, e.g. Jewett (2015) showed that neurons express TNF and IL1 in response to optogenetic stimulation. IL1 and TNF are well-characterized sleep regulatory and immune regulatory molecules.

9. Most, if not all, well-characterized sleep regulatory molecules are activity dependent, locally produced/released, influence cerebral blood flow, brain inflammation, brain plasticity, metabolism, and sleep locally. These molecules include adenosine, ATP, NO, prostaglandins, IL1, TNF, and BDNF. This is not coincidence; how these central actions influence and are separated from systemic actions leading to pathology is the central point of the review and thus could be folded into the review.

10. Line 153: "fever causes sleep..." No, it does not. There are many molecules that cause fever but not sleep. During bacterial infections sleep responses are independent of fever responses. E.g., see Toth et al (Infect & Immunity) of what happens to sleep during bacteremia. Sleep has a biphasic response during the first 24 hours, first increasing then later decreasing below control values while fever develops during that time as stays above control values for days.

Minor concerns:

11. Third paragraph, lines 51-57 is all one sentence. Break it up into several sentences for ease of reading.

12. The role NFkB has in sleep was first developed in the 1990's.
13. Toth's work with rabbits and trypanosomes is a great example showing how the immune status of an animal affects sleep. In rabbits, trypanosomes undergo an antigenic shift every 21 days; when that happens there is a sleep response every 21 days.
14. Lines 205-209; this reviewer disagrees with ever generalization made in this paragraph. Since there is not consensus this paragraph could be omitted.
15. Line 239; there is much evidence suggesting a complex, but tightly interwoven regulatory mechanisms between homeostatic and circadian sleep mechanisms. Fontana's work could be mention in this paragraph.
16. Line 544; since GH is mentioned, it might be worth mentioning that the sleep inducing actions of IL1 are dependent upon GHRH receptors and that both IL1 and GHRH act on hypothalamic GABAergic neurons.
17. Lines 819-828; The sleep link to AD-related inflammation is mostly speculative.
18. Figures 1 and 2 could be modified to illustrate the brain cytokine network including expression of anti-inflammatory members of various cytokine families.

RESPONSE TO REVIEWERS

Referee 1 comments	Replies
General comment: The review is overall well written and covers an important and current topic that is interesting to a broad, interdisciplinary research community. It could be structured somewhat better, because it touches several broad topics but remains superficial in the discussion of certain areas. To improve this, I would suggest to keep the focus more on the effects of sleep deprivation on disease outcome, which is already an extensive and highly relevant topic that so far has been less covered in publications within the sleep-immune research field. This would also help to reduce the strong overlap with several recent publications that have extensively reviewed basic sleep-immune interactions (Annu Rev Psychol. 2015, PMID: 25061767, Physiol Rev. 2019, PMID: 30920354, Nature Reviews Immun. 2019, PMID: 31289370).	We thank the reviewer for his/her appreciation for our work and for the valuable suggestions he/she has given us. We agree with the reviewer on the need for a more focused attention to the effects of sleep deprivation on disease outcomes, which was actually our real intention in the manuscript against a literature background mainly addressing basic mechanisms linking sleep loss and immune deregulation. Due to space limitation and in accordance with the referee suggestion, we have cut by more than half the length of the paper and reorganized its structure, reducing overlap with previous reviews and mainly focusing on and expanding risk and outcomes of sleep deprivation. We have addressed each concern point-by-point as follows, and changes are tracked in the paper to indicate the improvement accordingly.
1) Although the manuscript is overall well written, some sentences are quite long or difficult to understand. Some examples are: 1.1 Abstract, line 28f: Why would a “further understanding of the causal relationship between SD ... help ... to improve measures of sleep hygiene”? 1.2 Lines 47-50: “However, in the modern hyper-technological era we are witnessing an impressive increase in chronic sleep problems, that are closely linked to the growing and unstoppable pressure on sleep leading to the development of a constant sleep debt or insufficient sleep, likewise referred to as sleep deprivation (SD).”: The phrasing could be simplified. 1.3 Lines 51-57: “Nowadays, apart from medical problems, including obstructive sleep apnea (OSA), insomnia and others, factors associated with the modern 24/7 society such as globalization, societal demand for constant activities, increasing working time and schedule with more frequent shift-work, educational demands, internet explosion and communication/information technology as well as poor diet and drugs, contribute to a misalignment between the biological necessity of good sleep and the intense daily activities and lifestyle, thus changing	We are grateful for these comments that allow us to correct the text. 1.1 Thank you for this note. We have corrected the sentence as follows (lines 25-26): “Further understanding of the causal relationship between SD and immune deregulation would help to identify individuals at risk for disease and to prevent adverse health outcomes”. 1.2 Thank you for this note. Due to the word reduction and rephrasing of the introduction, we have cut the sentence in the revised manuscript. 1.3 Thanks for this note. The sentence has been simplified as follows (lines 47-50): “Besides medical problems, including obstructive sleep apnea (OSA), insomnia, factors mostly associated with the modern 24/7 society including work and social demands, smartphone addiction, poor diet, contribute to cause the current phenomenon of chronic sleep deprivation (SD), that is sleeping less than the recommended amount or, better to say, the intrinsic sleep need”.

sleep duration patterns and causing the currently pervasive phenomenon of self-induced chronic SD, that is sleeping less than the recommended amount or, better to say, the intrinsic sleep need⁴.” This is an extremely long sentence that could be shortened. 1.4 Lines 83ff: “...sleep hygiene is becoming an increasingly urgent public issue for both the scientific and medical communities in order to promote more awareness and education on sleep at the population and individual levels and hence prevent or attenuate the adverse health outcomes associated with insufficient sleep.” What does the part “in order to promote” refer to?	1.4 Thanks for this note. This sentence has been deleted from the introduction due to space limitation.
2) Lines 373ff: “Indeed, compared with regular nocturnal sleep, acute and mostly sustained SD has been found: a) to alter circulating leukocyte counts with studies reporting increased numbers of total leukocytes and specific cell subsets mainly neutrophils, monocytes, B cells, decreased circulating NK cells, and decreased circulating CD4+ T cells, in comparison with undisturbed sleep which is associated with low leukocyte counts, possibly due to a redistribution of leukocyte subsets to lymphoid organs where they participate in innate and adaptive immunity^{84,87-90}”; This sentence is somewhat redundant. In addition, circulating CD4+ T cells were found to be increase rather than decreased by SD.	Thank you for this important note. We have modified the sentence to remove the redundancy. Regarding the CD4+ T cell subset (line 464-467), we have corrected the typo.
3) The title of the review should more accurately reflect the focus of the manuscript on the role of insufficient sleep in disease risk and outcome.	We gladly accept the reviewer’s suggestion, having now focused the paper more on health outcomes of sleep deprivation. We have changed the title in: “Role of sleep deprivation in immune-related disease risk and outcomes”.
4) Some statements lack references, e.g. - Lines 42-44 - Lines 97-103 - Lines 343-346 - Lines 356-359 - Lines 562-566	References have been added.
5) Lines 113-116: The Introduction ends with the statement “Here, we reviewed the evidence regarding the role of chronic SD and the potential pathogenic mechanisms in the dysregulation of the immune function and the induction of unbalanced pro-inflammatory responses, as a risk factor for the development and/or progression of several immune-related chronic diseases. “, yet the next sections also deal with several other aspects, such as the regulation of sleep by immune products and circadian factors. The relevance of these topics for the main focus of the review should either be clarified more or the review should be more focused on directly relevant topics.	We appreciate this suggestion and, accordingly, we have rephrased the sentence in order to more clearly explain the aim of our review. Furthermore, we have strongly reduced aspects related to basic mechanisms and focused more on clinical outcomes. Please see lines 183-185.

6) Lines 196 – 204: The description of sleep stages seems misplaced here. The authors may consider giving a broader introduction to the basics of sleep or remove this description entirely.	We thank for this suggestion. We have removed the description of sleep stages.
7) In general, the review would benefit from including also some critical findings with opposite results and from a more in-depth discussion of the findings. These are only some examples: 7.1 Lines 494-497: There is a lot of evidence that β-adrenergic signaling is anti-inflammatory whereas α-adrenergic signaling is pro-inflammatory (see for example PMID: 25703786). 7.2 Lines 595ff: “Moreover, the genomic response to acute total SD is strongly intensified after chronic SD than after sufficient sleep, with more up-regulated genes belonging to IL-6 signaling, inflammatory and defense responses, phagocytosis, response to external stimuli and wounding, suggesting an exacerbated response to stress and challenges and predisposing to negative health outcomes⁵⁶.”: The conclusion that an up-regulation of genes involved in defense responses and phagocytosis predisposes to negative health outcomes is very speculative. One could likewise argue in the opposite way. 7.3 Lines 1047ff: “In parallel, cognitive behavior therapy reduced the stimulated monocyte production of IL-6 and TNF-α and the expression of pro-inflammatory transcripts.”: This finding appears to contradict studies that found an increasing effect of sleep on the stimulated production of IL-6 and TNF-α. This could be discussed.	We agree with the reviewer on the importance of including contrasting and expanding results. With regard to the immune response to sympathetic activation, we have now included a more in-depth discussion on adrenergic receptor involvement and related inflammatory outcomes. Please see lines 1050-1061. Fundamentally, we agree with the reviewer and thus we have avoided any speculations. However, due to space limitation and greater focus on clinical outcomes, the specific concept has now been removed. According to the reviewer’s note, we have now expanded this point in a more pertinent position of the paper (please see lines 475-481), where the effects of regular sleep compared with sleep loss on immune parameters are discussed.
8) Lines 521-522: “Normally, cortisol suppresses the pro-inflammatory gene expression (predominantly via mineral corticoid receptor) and upregulates the production of anti-inflammatory cytokines¹⁵³”: Please check whether this statement is correct. Many effects of cortisol on pro-inflammatory signaling are mediated via glucocorticoid not mineralocorticoid receptors.	We acknowledge this typing mistake and thank the reviewer for highlighting it. However, in the revised manuscript because of space limitation and greater focus on clinical outcomes, the specific statement has been now removed.
9) Lines 688-689: The first sentence is not well connected to the rest of the paragraph.	In accordance with this comment, the first sentence has been deleted and the entire paragraph rearranged, please see lines 1332 and further on.
10) Fig. 1: The line indicating an inhibition of T cells is somewhat confusing. From the depiction in the figure, it is not clear whether SD increases or decreases numbers of Th1 and Th2 cells.	Thanks for this comment. We have modified the Figure 1 now being Figure 2, showing separately the negative influence (line) of SD on T cells antigen presentation, on Th1 differentiation, and the

	predominance of Th2 differentiation (arrow), as observed in many studies discussed in the manuscript.
11) Lines 699-701: "... demonstrate that both the memory phase, indexed by the antigen-specific antibody response, and the effector phase, marketed by T cell number and cytokine production ... ". This statement is not correct. Effector and memory phases are both marked by antigen-specific antibody and T cell responses.	We agree with the reviewer on this note. We acknowledge that both immunological phases are characterized by antigen-specific antibody and T cell responses, thus the sentence has been modified as follows (lines 1369-1373): "Studies in which SD (one or few nights) was applied to healthy humans during (mostly after) the immunological challenge of vaccination demonstrate that both the memory and effector phases of the immune response, indexed by the antigen-specific antibody response and T cell number and cytokine production, are supported by sleep, that doubles the response to vaccination and seems to promote clinical protection".
12) Table 1: It would be interesting to show the effects of recovery sleep not only compared with changes after SD but also compared with baseline measurements (i.e., before sleep manipulation), to see whether recovery sleep entirely compensates the effects of SD.	This is an interesting note, that has allowed us to remodulate the Table 1, in order to more accurately present the results of the studies, showing where the immune effects of recovery sleep were statistically compared with baseline and/or with SD.
13) Lines 982-983: The statement "Hence, the ... pro-inflammatory effect of short sleep duration is reversible by increasing habitual sleep duration" should be toned down, because the effect was not significant.	We thank for this note and agree with it. However, because of space limitations, we have been forced to remove the specific statement.
14) Lines 1057ff: "Sleep exerts immunosupportive and inflammation-regulatory functions, and impairments of immune inflammatory system is a plausible mechanism mediating the negative health effects of SD, which leads to a response by the host immune defense similar to that of threats (infection or injury).": I don't think it is correct to say that SD leads to an immune response similar to that of infection or injury.	In agreement with the reviewer's comment, we have removed the overstatement and rephrased the sentence. Please see lines 1997-1999.
15) The manuscript should be carefully screened for grammar errors, e.g., line 87 (plays should read play), line 563 ("are thought the cellular line" should probably read "are thought to be..."), line 628 ("these evidence" should read "this evidence"), line 968 ("Whether SD or habitual short sleep represent...": whether should probably read if), etc.	We thank the reviewer for this note. We have checked the manuscript for grammar errors, and corrected them.
16) The word "subject", is sometimes not used appropriately, e.g. line 1055: "Epidemiological data clearly suggest that increasing numbers of subjects are becoming sleep deprived...".	We agree with the reviewer's comment. We have changed the word "subjects" in pertinent points.
Referee 2 comments	Replies
General comment: This manuscript is a clearly written comprehensive summary of the multiple effects of sleep loss (sleep	We sincerely thank the reviewer for the appreciation of our work and mostly for the

deprivation [SD]) on immune system related pathologies. In general, citations are appropriate and provide a reasonable guide for readers to go into greater depth. The emphasis on the effects of SD on specific pathologies/syndromes and relating those effects to changes in immune responses is well organized and appropriate. While it may not be unique, it is more focused on the SD-immune links in pathologies than prior reviews in this field, some of which are stellar, e.g. ref 34. Regardless, a few issues were identified. Herein they are organized into major concerns, suggested issue to address, and minor issues.	constructive critiques that have allowed us to improve the manuscript. Due to space limitation, we have cut by more than half the length of the paper (originally > 14.000 words, now around 6000 words) and reorganized its structure leaving the focus on clinical outcomes, therefore, regrettably, some specific paragraphs and/or concepts have now been removed, or addressed only concisely, and might include some of your comments. We have addressed each concern point-by-point as follows, and changes are tracked in the paper to indicate the improvement accordingly.
1) The brain network of cytokines, including the cells that express them, what stimulates them, their involvement in multiple brain functions, e.g. synaptic scaling, brain development, sleep of course, appetite, mental health status etc. is hardly mentioned. Further, the complexity of the brain cytokine network as it applies to sleep is not mentioned. For instance, within the IL1 and TNF families of molecules there are more anti-inflammatory members than pro-inflammatory members. Further, the effects of IL1 and TNF on sleep are dose-dependent – low doses promote sleep while high doses inhibit sleep – and time of day dependent. Some of the brain cytokine molecules are specific to brain. E.g. the IL1 receptor accessory protein (AcP) has 3 isoforms, one is specific to neurons (called AcPb); AcPb, [not AcP or the soluble form of AcP], upregulates with sleep loss and is required for sleep rebound to occur after sleep loss. AcPb is anti-inflammatory. Other examples illustrating the nuances of the brain cytokine network include the observations that forward TNF signaling (i.e. soluble TNF binding to one of its transmembrane receptors) promotes sleep while reverse TNF signaling (the soluble TNF receptor binding to the transmembrane form of TNF) promotes waking. Finally, both IL1 and TNF have daily rhythms of expression in brain with peaks occurring at ZT0, time of sleep period onset in rats and mice.	According to this important and useful suggestion, we have now examined in depth aspects related to the interesting brain cytokine network within the Section entitled “Basic immune mechanisms of sleep regulation” please lines 187; 197-202; 419-446.
2) How systemic cytokines, which are the major focus of human studies of SD-immunity, relate to brain cytokine expression and actions are not developed adequately. E.g. there is a literature indicating that systemic cytokines induce action potentials in vagal afferents which in turn induce brain cytokine mRNA production. Such effects are lost with vagotomy. Similar studies show how bacterial products also are dependent in part on vagal afferents to affect brain cytokine levels. For low grade inflammation, this is likely a major way of influencing the brain cytokine orchestra and many of the SD-linked pathology outcomes such as the autonomic parasympathetic and sympathetic nervous systems influence as was	According to this interesting suggestion, in the paragraph entitled “Basic immune mechanisms of sleep regulation” (line 187) we have discussed the interrelation between brain and systemic cytokine circuits. We have also introduced a related new Figure 1.

developed in the review. Given the large gap between animal studies, within which brains can be examined, and human studies, which are mostly restricted to systemic sampling, these mechanisms need to be addressed as they provide validity to the human studies.	
3) There are known links between cytokines and circadian rhythm molecular mechanisms. Perhaps most important for this review are those developed by A. Fontana's lab in Zurich where it was shown that TNF and IL1 directly affect clock gene expressions. This could be mentioned in multiple places in the manuscript.	According to the reviewer's comment, we have introduced these important data: please see lines 457-462.
4) In contrast, the authors devote a few paragraphs to the two-process model of sleep and a related triple-drive model. These are mathematical models which are not concerned with molecular mechanisms and compartmentalize sleep regulation into two or three components for which we already know complex interactions, e.g. see reference to Fontana's work above. The models although useful in various ways, are not helpful to link SD to immunity and pathology. Further, the primary outcome measure used to quantify process S, EEG slow wave activity, is regulated independently of sleep duration and does not reflect sleep intensity in multiple conditions, e.g. neonates and the elderly mostly lack EEG SWA, some sleep pills inhibit EEG SWA while increasing sleep, etc. The paragraphs discussing mathematical models could be eliminated without loss meaningful manuscript content.	In accordance with the reviewer's suggestion and due to space limitation, we have removed the paragraph "Sleep-wake regulation: the synergy between sleep and circadian drives", which included the mathematical models of sleep, and reorganized basic concepts in the new paragraph entitled "Basic immune mechanisms of sleep regulation" (line 187).
5) Somewhere in the introduction the simple observations that almost all physiological measures, be it hormone levels, blood CO2 levels, mentation, heart rate, respiratory rate, locomotor activity, etc. are different between sleep and wake states. This coupled with the effects of SD on genome-wide expression of transcripts, which indicate that a very large number of genes are altered by SD, over 50% in some studies, suggest limitations on interpretation of the effects of SD to the extent that the effects observed are the result of sleep loss per se. These broad changes between sleep and wake indicate that few, if any, studies have ever isolated sleep as the independent variable; as such causality is not possible to identify, thereby limiting conclusions.	We agree with the reviewer on this note. Concordantly, we have introduced this limitation at lines 2000-2003, where the following sentence has been added: "Caution should be exercised in interpreting cellular and molecular outcomes of SD in experimental studies conducted till now as a result of an independent effect of SD, because other factors may play a role, including extended wakefulness-associated processes, other features of sleep-wakefulness, their temporal and functional segregation or methodologies of sleep manipulation".
6) The paragraph (lines 169-176) within which hypotheses dealing with sleep function is insufficiently develop and could be omitted since none of the hypotheses mentioned are universally accepted nor are they relevant to the immune system with the possible exception of the idea the sleep removes toxic biomolecules from the brain. That idea, while very attractive, is mostly based on experimental	We appreciate this note. The paragraph has been removed, and aspects related to synaptic plasticity has been now addressed with reference to cytokines role, please see lines 426-429 of the new paragraph entitled "Basic immune mechanisms of sleep regulation" (line 187).

evidence gathered during states of anesthesia. The ideas of Obal and Krueger (1993) and Kavanau (1995, 1996) positing that sleep is required for preserving, or stabilizing, synaptic plasticity, and is activity-dependent, are not even mentioned even though TNF is a well-characterized brain molecule tied to synaptic scaling. Thus, the only theories of sleep function directly tied to “immune molecules” is not mentioned.	
7) Lines 72-92; It may be interesting to address the multiple findings relating sleep and sleep loss to mental illness which in turn can affect immune function.	We have introduced relevant findings on the link between sleep deprivation and mental illness, please see lines 66-68.
8) Line 176; the concept of local use-dependent sleep was first put forward by Obal and Krueger in 1993 and they and others have developed the story in the intervening years, perhaps most convincingly by D. Rector. Some of that work is directly related to the subject of this review, e.g. Jewett (2015) showed that neurons express TNF and IL1 in response to optogenetic stimulation. IL1 and TNF are well-characterized sleep regulatory and immune regulatory molecules.	In agreement with this comment, we have addressed these interesting points in the paragraph entitled “Basic immune mechanisms of sleep regulation” (line 187), please specifically see lines 430-446.
9) Most, if not all, well-characterized sleep regulatory molecules are activity dependent, locally produced/released, influence cerebral blood flow, brain inflammation, brain plasticity, metabolism, and sleep locally. These molecules include adenosine, ATP, NO, prostaglandins, IL1, TNF, and BDNF. This is not coincidence; how these central actions influence and are separated from systemic actions leading to pathology is the central point of the review and thus could be folded into the review.	In agreement with this comment, we have addressed these interesting points in the paragraph entitled “Basic immune mechanisms of sleep regulation” (line 187).
10) Line 153: “fever causes sleep...” No, it does not. There are many molecules that cause fever but not sleep. During bacterial infections sleep responses are independent of fever responses. E.g., see Toth et al (Infect & Immunity) of what happens to sleep during bacteremia. Sleep has a biphasic response during the first 24 hours, first increasing then later decreasing below control values while fever develops during that time as stays above control values for days.	In agreement with this note, we have modified the sentence and added relevant references, please lines 451-456.
11) Third paragraph, lines 51-57 is all one sentence. Break it up into several sentences for ease of reading.	We have simplified the sentence as follows (lines 47-50): “Besides medical problems, including obstructive sleep apnea (OSA), insomnia, factors mostly associated with the modern 24/7 society including work and social demands, smartphone addiction, poor diet, contribute to cause the current phenomenon of chronic sleep deprivation (SD), that is sleeping less than the recommended amount or, better to say, the intrinsic sleep need”.
12) The role NFkB has in sleep was first developed in the 1990’s.	Thanks for this note. We have reported it at lines 1069-1070.

13) Toth's work with rabbits and trypanosomes is a great example showing how the immune status of an animal affects sleep. In rabbits, trypanosomes undergo an antigenic shift every 21 days; when that happens there is a sleep response every 21 days.	We have expanded these interesting supportive findings at lines 453-455.
14) Lines 205-209; this reviewer disagrees with ever generalization made in this paragraph. Since there is not consensus this paragraph could be omitted.	We agree, and thus we have removed the paragraph.
15) Line 239; there is much evidence suggesting a complex, but tightly interwoven regulatory mechanisms between homeostatic and circadian sleep mechanisms. Fontana's work could be mention in this paragraph.	According to the reviewer's comment, we have introduced these important data: please see lines 457-462.
16) Line 544; since GH is mentioned, it might be worth mentioning that the sleep inducing actions of IL1 are dependent upon GHRH receptors and that both IL1 and GHRH act on hypothalamic GABAergic neurons.	We thank for this issue, that has been added at lines 200-202.
17) Lines 819-828; The sleep link to AD-related inflammation is mostly speculative.	We agree with the reviewer's suggestion. Congruently, the following sentence has been added at lines 1678-1679: "However, the relationship of SD to inflammation in AD is mostly speculative and needs to be confirmed".
18) Figures 1 and 2 could be modified to illustrate the brain cytokine network including expression of anti-inflammatory members of various cytokine families.	According to the suggestion, a novel separate Figure 1 has now been produced to help visualize the brain cytokine network and its systemic interaction.

Reviewers' comments:

Reviewer #1 (Remarks to the Author):

The revision improved the manuscript somewhat, but at some places it is still difficult to follow the authors' thoughts and the discussion of findings is, in part, still quite superficial. It was very difficult to see the single responses to my comments, because some of the mentioned lines numbers were either missing or were completely wrong (even in the new version of the response letter). Since the first part of the manuscript was substantially edited, I read it as a new manuscript. See below my specific comments. Please note that the line numbers mentioned here refer to the manuscript version with the tracked changes.

- The authors should consider moving the chapter "Sleep deprivation-induced immune dysregulation" after discussion of the literature on sleep and diseases. The chapter could then focus on potential mechanisms underlying the association between poor sleep and diseases, which is the main focus of the review. At the moment, the first two chapters appear a bit misplaced.

- Several sentences are (still) not well-phrased or difficult to understand. E.g.:

- Lines 47-50: "Besides medical problems, including obstructive sleep apnea (OSA), insomnia, factors mostly associated with the modern 24/7 ... "

- Lines 173-175: "Links exist between IL-1 β and GHRH/GH in promoting sleep so that IL-1-induced growth hormone (GH) release via GHRH55, and hypothalamic gamma-aminobutyric acid (GABA)ergic neurons (promoting sleep) are responsive to both GHRH and IL-1 β ".

- Lines 1460ff: "Studies in which SD (one or few nights) was applied to healthy humans during (mostly after) the immunological challenge of vaccination demonstrate that both the memory and effector phases of the immune response, indexed by the antigen-specific antibody response and T cell number and cytokine production, are supported by sleep, that doubles the response to vaccination and seems to promote clinical protection¹⁹⁰."

- Lines 1629f: "SD is associated with a rapid decline in circulatory melatonin levels, linked to rapid consumption of melatonin as a first-line defense against the associated rise in oxidative stress²²⁸."

- Lines 176-179: This paragraph does not fit well here. It starts with anti-inflammatory cytokines and then focusses again on proinflammatory cytokines, which have been discussed already in the paragraph before.

- Line 389: What are these TNF and IL-1 TNF families?

- Lines 403-404: "Among these, a sleep response is induced thereby favoring recovery from infection and inflammation, via the timely functional investment of energy into the energy consuming immune processes^{58,94}." This is a speculation and should be clearly indicated as such.

- In my opinion Figure 2 is too simplified without giving much information. Some parts are also not clear. E.g., what do the arrows to the lymph node indicate? Why does the figure suggest that sleep deprivation increases antibody production in B cells? How do the immunological changes relate to the diseases shown?

- Lines 419ff: The enumeration is too simplified as sleep deprivation has also been shown to induce opposite effects to the ones mentioned.

- Lines 429-435: This part is not well structured. Some findings are reported twice and the authors jump from one cytokine to another and back. Also, some studies found increases in TNF production while others found decreases. These contradicting findings are mentioned but not discussed. Please also make sure to correctly indicate whether changes in immune parameters were observed between nocturnal sleep versus nocturnal wakefulness or night time vs. daytime. This shouldn't be mixed up

(like in this sentence "Other studies found nocturnal increased TNF- α production by LPS-stimulated monocytes compared with daytime¹²⁰".

- Lines 436f: "Undisturbed sleep is predominantly characterized by a Th1 polarization of T helper cells (expressing IFN- γ , IL-2 and TNF- α), but experimental SD in humans leads to a shift from a Th1 pattern towards a Th2 pattern (expressing IL-4, IL-5, IL-10, and IL-13)^{118,123,438}".: I would not use the word "but" here as the findings do not contradict each other.

- Lines 1133-1144: The role of catecholamines is still not very clear. They can mediate anti- as well as pro-inflammatory effects, but what is the "net" effect that can be expected? Could this be discussed or is the literature not clear? If so, this should be mentioned.

- Lines 1416-1418: This sentence does not fit well here.

- Lines 1752f: "In a recent prospective cohort, a low-stable sleep pattern (<5 h sleep/night) during the 4-y follow-up had the highest risk of death and CV events, highlighting also the importance of the temporal rather than static behavior of sleep duration²⁵⁵".: What is meant with temporal and static behavior of sleep duration?

- Figure legend to Fig. 2: "Sleep deprivation, as induced experimentally or in the context of habitual short sleep, leads to...": Claims of causality should be avoided when referring to observational studies.

- Fig. 3 suggests that growth hormone and prolactin inhibit telomere shortening. Is there any evidence for this?

- The term "subject(s)" is still used wrongly at several places in the manuscript.

Reviewer #2 (Remarks to the Author):

The revised manuscript is much improved. It is now focused and much easier to read.

There are still some minor typo's - my computer caught most of them so it will be easy to correct them and I do not list them.

My only suggestion is that Figs 2 and 3 should show brain production of cytokines and their release into the circulation. Harvey Moldofsky (Univ. of Toronto) years ago showed that TNF in the brain reaches the circulation. Many others, and many of those are cited, have shown brain production of cytokines.

RESPONSE TO REVIEWERS

Referee 1 comments	Replies
The revision improved the manuscript somewhat, but at some places it is still difficult to follow the authors' thoughts and the discussion of findings is, in part, still quite superficial. It was very difficult to see the single responses to my comments, because some of the mentioned lines numbers were either missing or were completely wrong (even in the new version of the response letter). Since the first part of the manuscript was substantially edited, I read it as a new manuscript. See below my specific comments. Please note that the line numbers mentioned here refer to the manuscript version with the tracked changes.	We agree with the reviewer on the many differences in comparison with the original version of the manuscript. According to the first round revision, we had to introduce some new concepts and to change many parts of the text, also due to text reduction in order to comply with the editorial guidelines on word limit. We sincerely apologize for this. Moreover, there might have been pitfalls with the Microsoft Word track changes and the corresponding line numbers that we could not explain. We thank the reviewer's for his/her remarks, that allow for paper improvements. In the present revised version, we have addressed the reviewer's issues as reported in the text in track changes and highlighted in yellow. In the following point-by-point responses, please refer to the line numbers of the Word file with changes displayed "All Markup" and comments shown in the right balloons.
The authors should consider moving the chapter "Sleep deprivation-induced immune dysregulation" after discussion of the literature on sleep and diseases. The chapter could then focus on potential mechanisms underlying the association between poor sleep and diseases, which is the main focus of the review. At the moment, the first two chapters appear a bit misplaced.	We agree with this valuable remark. Accordingly, the chapters order has been switched so that now the discussion starts with the literature on SD and disease risk and outcome ("Sleep deprivation and immune-related disease outcomes"), and then deals with the immune mechanisms proposed to be a link between SD and diseases. To this purpose, this last chapter has changed the title from "Sleep deprivation-induced immune dysregulation" into "Immune mechanisms linking sleep deprivation and diseases", because this seems to us more focused and in line with the previous chapter.
Several sentences are (still) not well-phrased or difficult to understand. E.g.:	
- Lines 47-50: "Besides medical problems, including obstructive sleep apnea (OSA), insomnia, factors mostly associated with the modern 24/7 ... "	The sentence has been corrected as follows (lines 49-50): "Besides medical problems including obstructive sleep apnea (OSA) and insomnia, factors associated mostly with the modern 24/7 society such as work and social demands, smartphone addiction, and poor diet,...".
Lines 173-175: "Links exist between IL-1β and GHRH/GH in promoting sleep so that IL-1-induced growth hormone (GH) release via GHRH55, and hypothalamic gamma-aminobutyric acid (GABA)ergic neurons (promoting sleep) are responsive to both GHRH and IL-1β".	The sentence has been corrected as follows (lines 230-233): "Links exist between IL-1β and GHRH/GH in promoting sleep so that IL-1 induced growth hormone (GH) release via GHRH, and hypothalamic gamma-aminobutyric acid (GABA)ergic neurons (promoting sleep) are responsive to both GHRH and IL-1β".
- Lines 1460ff: "Studies in which SD (one or few nights) was applied to healthy humans during (mostly after) the immunological challenge of vaccination demonstrate that both the memory and	The sentence has been modified as follows (lines 831-834): "Studies in which SD (one or few nights) was applied to healthy humans during (mostly after) the immunological challenge of vaccination

effector phases of the immune response, indexed by the antigen-specific antibody response and T cell number and cytokine production, are supported by sleep, that doubles the response to vaccination and seems to promote clinical protection¹⁹⁰.”	demonstrate that SD reduced both the memory and effector phases of the immune response, as indexed by suppressed antigen-specific antibody and Th cell response compared with undisturbed sleep.”
- Lines 1629f: “SD is associated with a rapid decline in circulatory melatonin levels, linked to rapid consumption of melatonin as a first-line defense against the associated rise in oxidative stress²²⁸.”	The sentence has been modified as follows (lines 1135-1136): “SD is associated with a rapid decline in circulatory melatonin levels, which may be linked to rapid consumption of melatonin as a first-line defense against the SD-associated rise in oxidative stress”.
Lines 176-179: This paragraph does not fit well here. It starts with anti-inflammatory cytokines and then focusses again on proinflammatory cytokines, which have been discussed already in the paragraph before.	We thank the reviewer for this valuable suggestion. In accordance, the paragraph has been modified to avoid redundancy and reorganize it in a more coherent discussion (please see lines 227-234).
Line 389: What are these TNF and IL-1 TNF families?	The sentence has been modified as follows (line 612): “TNF and IL-1 are closely interconnected and play a similar role in the regulation of sleep”.
- Lines 403-404: “Among these, a sleep response is induced thereby favoring recovery from infection and inflammation, via the timely functional investment of energy into the energy consuming immune processes^{58,94}.”: This is a speculation and should be clearly indicated as such.	Thanks for the suggestion. We have now rephrased to clearly highlight that is only a speculation. Please see lines 626-632): “Among these, a sleep response is induced and has been hypothesized to favor recovery from infection and inflammation, supposedly via the timely functional investment of energy into the energy consuming immune processes.”
In my opinion Figure 2 is too simplified without giving much information. Some parts are also not clear. E.g., what do the arrows to the lymph node indicate? Why does the figure suggest that sleep deprivation increases antibody production in B cells? How do the immunological changes relate to the diseases shown?	Accordingly, we have remodulated the Figure 2 in order to report immune mechanisms which are altered by SD and are suggested or found to be involved in the pathogenesis of immune-related diseases associated with SD.
Lines 419ff: The enumeration is too simplified as sleep deprivation has also been shown to induce opposite effects to the ones mentioned.	Thank you for pointing this out. We have restructured the part eliminating the enumeration and also introducing some opposite findings. Please see lines 1411-1423.
Lines 429-435: This part is not well structured. Some findings are reported twice and the authors jump from one cytokine to another and back. Also, some studies found increases in TNF production while others found decreases. These contradicting findings are mentioned but not discussed. Please also make sure to correctly indicate whether changes in immune parameters were observed between nocturnal sleep versus nocturnal wakefulness or night time vs. daytime. This shouldn't be mixed up (like in this sentence “Other studies found nocturnal increased TNF-α production by LPS-stimulated monocytes compared with daytime¹²⁰”.	Thanking the reviewer for this important remark, the part has now been checked and restructured in accordance with the suggestions. Please see lines 1433-1439: “Regarding cellular markers of inflammation, some studies found that the ex-vivo LPS-stimulated production of TNF-α^{232,233}, IL-1β and IL-6^{203,232-234} by human monocytes increased during SD but decreased during regular nocturnal sleep^{54,203,232-234}. However, other studies reported a decrease of TNF-α production by activated monocytes after SD compared with regular nocturnal sleep^{203,235}. These contrasting results need further investigations and may depend on differences in the cytokine sensitivity to different SD protocols or sampling methods and time. For instance, it seems that partial acute SD increased

	stimulated monocytic TNF- α production ^{232,233} , while more sustained SD decreased it ^{203,235} .
Lines 436f: “Undisturbed sleep is predominantly characterized by a Th1 polarization of T helper cells (expressing IFN-g, IL-2 and TNF-a), but experimental SD in humans leads to a shift from a Th1 pattern towards a Th2 pattern (expressing IL-4, IL-5, IL-10, and IL-13)118,123438”: I would not use the word “but” here as the findings do not contradict each other.	Concordantly, we have rephrased as follows (lines 1440-1442): “Undisturbed sleep is predominantly characterized by a Th1 polarization of T helper cells (expressing IFN- γ , IL-2 and TNF- α), and experimental SD in humans leads to a shift from a Th1 pattern towards a Th2 pattern (expressing IL-4, IL-5, IL-10, and IL-13).”
Lines 1133-1144: The role of catecholamines is still not very clear. They can mediate anti- as well as pro-inflammatory effects, but what is the “net” effect that can be expected? Could this be discussed or is the literature not clear? If so, this should be mentioned.	The immune role of sympathetic nervous system activation (SNS) by SD is not entirely clear despite most evidence point to a pro-inflammatory effects, albeit indirectly. This point has been evidenced in the paper. Recent studies as discussed in the paper (ref 255 and 256) are specifically performed in the context of SD, and suggest the involvement of SNS in the proinflammatory response associated with SD. Modifications in the paragraph addressing catecholamine role are highlighted: please see lines 1717-1719 and 1726-1728.
Lines 1416-1418: This sentence does not fit well here.	We agree. Considering that the sentence is misplaced and not crucial for the discussion, it has been deleted (please see line 657).
- Lines 1752f: “In a recent prospective cohort, a low-stable sleep pattern (<5 h sleep/night) during the 4-y follow-up had the highest risk of death and CV events, highlighting also the importance of the temporal rather than static behavior of sleep duration ²⁵⁵ .”: What is meant with temporal and static behavior of sleep duration?	Accordingly, we have rephrased as follows (lines 1233-1255), to avoid misunderstanding: “In a recent prospective cohort, a low-stable sleep pattern (<5 h sleep/night) during the 4-y follow-up had the highest risk of death and CV events”.
Figure legend to Fig. 2: “Sleep deprivation, as induced experimentally or in the context of habitual short sleep, leads to...”: Claims of causality should be avoided when referring to observational studies.	In agreement with the comment, the sentence in the Legend to Fig. 2 has been now modified to avoid speculation on causality, as follows: “Sleep deprivation, as induced experimentally or in the context of habitual short sleep, has been found to be associated with alterations in...”.
Fig. 3 suggests that growth hormone and prolactin inhibit telomere shortening. Is there any evidence for this?	No, there is no evidence that growth hormone and prolactin inhibit telomere shortening. As such, we have deleted telomere attrition in the Figure 3.
The term “subject(s)” is still used wrongly at several places in the manuscript.	We agree. The text has been re-checked for wrong use of the term “subject(s)” and corrected pertinently, using instead, for instance, “participants” or “patients” or others.
Referee 2 comments	Replies
The revised manuscript is much improved. It is now focused and much easier to read. There are still some minor typo's - my computer caught most of them so it will be easy to correct them and I do not list them.	We are sincerely grateful to the reviewer for the insightful comments and valuable improvements to our paper. The revised manuscript has been checked and corrected for typos.
My only suggestion is that Figs 2 and 3 should show brain production of cytokines and their	As suggested by the reviewer, the brain production of cytokines and their systemic release have now

release into the circulation. Harvey Moldofsky (Univ. of Toronto) years ago showed that TNF in the brain reaches the circulation. Many others, and many of those are cited, have shown brain production of cytokines.

been included in Figure 2 and Figure 3.